# Molecular consequences of peripheral Influenza A infection on cell populations in the murine hypothalamus

**René Lemcke[1]\*, Christine Egebjerg[1], Nicolai T Berendtsen[1], Kristoffer L Egerod[2], Allan R Thomsen[3], Tune H Pers[2], Jan P Christensen[3], Birgitte R Kornum[1]\***

[1]Department of Neuroscience, Faculty of Health and Medical Sciences, University of Copenhagen, Copenhagen, Denmark; [2]Novo Nordisk Foundation Center for Basic Metabolic Research, Faculty of Health and Medical Sciences, University of Copenhagen, Copenhagen, Denmark; [3]Department of Immunology and Microbiology, Faculty of Health and Medical Sciences, University of Copenhagen, Copenhagen, Denmark

**\*For correspondence:**
rene.lemcke@gmail.com (RL);
kornum@sund.ku.dk (BRK)

**Competing interest:** The authors declare that no competing interests exist.

**Abstract** Infection with Influenza A virus (IAV) causes the well-known symptoms of the flu, including fever, loss of appetite, and excessive sleepiness. These responses, mediated by the brain, will normally disappear once the virus is cleared from the system, but a severe respiratory virus infection may cause long-lasting neurological disturbances. These include encephalitis lethargica and narcolepsy. The mechanisms behind such long lasting changes are unknown. The hypothalamus is a central regulator of the homeostatic response during a viral challenge. To gain insight into the neuronal and non-neuronal molecular changes during an IAV infection, we intranasally infected mice with an H1N1 virus and extracted the brain at different time points. Using single-nucleus RNA sequencing (snRNA-seq) of the hypothalamus, we identify transcriptional effects in all identified cell populations. The snRNA-seq data showed the most pronounced transcriptional response at 3 days past infection, with a strong downregulation of genes across all cell types. General immune processes were mainly impacted in microglia, the brain resident immune cells, where we found increased numbers of cells expressing pro-inflammatory gene networks. In addition, we found that most neuronal cell populations downregulated genes contributing to the energy homeostasis in mitochondria and protein translation in the cytosol, indicating potential reduced cellular and neuronal activity. This might be a preventive mechanism in neuronal cells to avoid intracellular viral replication and attack by phagocytosing cells. The change of microglia gene activity suggest that this is complemented by a shift in microglia activity to provide increased surveillance of their surroundings.

## eLife assessment

This **important** study combines experiments and computational approaches to understand the effects of influenza H1N1 infection on hypothalamic cells. The methodology and analysis are **solid** and raise questions around how a respiratory virus affects the central nervous system.

## Introduction

In humans as well as other species, Influenza A virus (IAV) infection causes respiratory tract disease, evoking seasonal epidemics associated with increased morbidity and mortality rates (*Macias et al., 2021*; *Paget et al., 2019*). Beyond seasonal epidemics, H1N1 IAV strains have been identified as

**eLife digest** When you are ill, your behaviour changes. You sleep more, eat less and are less likely to go out and be active. This behavioural change is called the 'sickness response' and is believed to help the immune system fight infection.

An area of the brain called the hypothalamus helps to regulate sleep and appetite. Previous research has shown that when humans are ill, the immune system sends signals to the hypothalamus, likely initiating the sickness response. However, it was not clear which brain cells in the hypothalamus are involved in the response and how long after infection the brain returns to its normal state.

To better understand the sickness response, Lemcke et al. infected mice with influenza then extracted and analysed brain tissue at different timepoints. The experiments showed that the major changes to gene expression in the hypothalamus early during an influenza infection are not happening in neurons – the cells in the brain that transmit electrical signals and usually control behaviour. Instead, it is cells called glia – which provide support and immune protection to the neurons – that change during infection. The findings suggest that these cells prepare to protect the neurons from influenza should the virus enter the brain.

Lemcke et al. also found that the brain takes a long time to go back to normal after an influenza infection. In infected mice, molecular changes in brain cells could be detected even after the influenza infection had been cleared from the respiratory system.

In the future, these findings may help to explain why some people take longer than others to fully recover from viral infections such as influenza and aid development of medications that speed up recovery.

---

the causal agent for the pandemics in 1918 and 2009, leading up to 50 million and 18,000 deaths worldwide, respectively (*Simonsen et al., 2013*; *Taubenberger and Morens, 2008*). Beyond respiratory tract disease, IAV infections have also been associated with neurological consequences such as encephalitis lethargica and narcolepsy. Evidence from the Spanish flu pandemic in 1918 linked H1N1 influenza infections to epidemic encephalitis lethargica (*Hoffman and Vilensky, 2017*; *Maurizi, 2010*). Similarly, increased numbers of encephalitis cases were reported during the 2009 H1N1 pandemic (*Goenka et al., 2014*; *Gu et al., 2013*; *Surana et al., 2011*). A peak in new narcolepsy cases was also seen following the 2009 H1N1 pandemic, and while this was linked with the Pandemrix vaccine in Europe, in China it was unrelated to vaccinations and thus might instead be linked to infection with the virus itself (*Ahmed et al., 2014*; *Han et al., 2011*). A better understanding of the direct and indirect effects of IAV infection on the brain is needed to shed light on these observations.

Some IAV strains are considered neurotropic, meaning that they hold the machinery to infiltrate the CNS and replicate, but evidence suggest that even non-neurotropic influenza strains can infiltrate the brain and cause neurological disturbances (*Ekstrand, 2012*; *Hodgson et al., 2012*; *Macias et al., 2021*; *Majde et al., 2007*; *Zielinski et al., 2013*). When non-neurotropic viruses are found in the CNS, they are either not able to replicate at all or not efficiently in the brain resident cells (*Hosseini et al., 2018*; *Zielinski et al., 2013*). The H1N1 strain from the 2009 pandemic (H1N1 pdm09) is non-neurotropic (*Sadasivan et al., 2015*), so the mechanism behind the neurological consequences seen after that H1N1 infection must be caused by indirect effects rather than direct viral infection of brain cells. It is well known that cytokines produced during a peripheral infection can reach the brain and cause functional disturbances (*Dantzer and Kelley, 2007*). It has also been shown that microglia become activated during peripheral IAV infections with non-neurotropic strains. Whether this microglia activation is a neuroprotective mechanism or contributes to neuroinflammation and -degeneration remains elusive (*Dusedau et al., 2021*; *Sadasivan et al., 2015*; *Wiley et al., 2015*).

The hypothalamus is a functional integrator of peripheral signals and regulates several physiological and behavioural functions via several specialised nuclei (*Berthoud and Münzberg, 2011*; *Bonnavion et al., 2016*; *Stuber and Wise, 2016*). The different hypothalamic neuron populations control among other things feeding, body temperature, sleep-wake states, and circadian rhythms (*Bonnavion et al., 2016*; *Timper and Brüning, 2017*; *Yousefvand and Hamidi, 2021*). The lateral hypothalamic area (LHA) contains well described neuron populations like hypocretin (HCRT) or melanin-concentrating hormone releasing (MCH) neurons. These neurons are coupled to several biological functions

including metabolism and the regulation of sleep and wake states (*Oesch and Adamantidis, 2021*). Recent studies showed that the LHA is also involved during inflammatory processes and regulates pain perception (*Bonnavion et al., 2016*; *Brown et al., 2015*; *Fakhoury et al., 2020*; *Stuber and Wise, 2016*). The dorsomedial hypothalamus (DMH) receives circadian input from the suprachiasmatic nucleus to regulate downstream behavioural and physiological changes including food intake, locomotion, and core body temperature (*Gooley et al., 2006*). Recent findings have linked neuronal subpopulations in the DMH to changes in thermoregulation and metabolism during immune activation with the bacterial cell wall component lipopolysaccharide (LPS; *Osterhout et al., 2022*). Several single-cell/nuclei-RNA sequencing (scRNA-seq, snRNA-seq) datasets have been published during the past years, shedding more light on the cell heterogeneity of the different hypothalamic regions, but little is known about how the transcriptome of these cell types change during environmental perturbations (*Campbell et al., 2017*; *Chen et al., 2017*; *Mickelsen et al., 2019*; *Mickelsen et al., 2020*; *Romanov et al., 2017*).

In the current study, we used intranasal infections with H1N1 pdm09 in female mice and snRNA-seq to follow changes in the transcriptional landscape of cell populations in the medial hypothalamus. We find changes in all populations during the peak of disease. A change in cell state is seen in microglia at early infection and later strong effects are detected in astrocytes and oligodendrocytes. Interestingly, after the mice have recovered from the acute infection, all non-neuronal cell types return to their resting state, while in neurons we detect long lasting changes of the transcriptome even after full recovery.

## Results

Mice were infected intranasally with influenza virus A/California/07/2009 (H1N1 pdm09) and monitored for a period of 23 days post infection for weight changes. Groups of mice were euthanized at 3, 7, and 23 days post infection (dpi) to collect lung, olfactory bulb (OB) and brain tissue for detection of viral abundance and snRNA-seq. The rationale for choosing these time points of sampling were as follows: (i) 3 dpi as a time point of initial disease onset before the start of weight loss – at this time point innate immunity is peaking; (ii) 7 dpi marks the point of strong weight loss and is where the adaptive immune response takes over, and before potential lethal outcome for some of the animals and (iii) 23 dpi as a recovery group where animals had completely regained their initial body weight (*Uddbäck et al., 2016*).

### IAV respiratory tract disease progression and viral infiltration into lung and olfactory bulbs

Infected mice started losing weight around 4 dpi and had the highest weight loss at 10 dpi (*Figure 1B*). Animals in the recovery group had regained their body weight at 23 dpi. Lungs and OB were tested for the abundance of the M1 gene of the virus, to test for viral infection of those tissues. Viral mRNA was detected at 3 and 7 dpi in the lungs but was not detected at 23 dpi (*Figure 1C*). In the OB, viral mRNA was detected in one animal at 3 dpi, in all animals at 7 dpi, and in four out of five animals at 23 dpi. Three of the animals did not loose weight due to the infection (7 dpi: 7.3; 23 dpi: 23.3 and 23.4) instead the presence of viral mRNA in lungs and/or OB confirmed a successful infection (*Figure 1— figure supplement 2*).

### Molecular and cellular composition of the medial hypothalamus snRNA-seq dataset

For snRNA-seq we dissected a hypothalamic region containing both the DMH and the LHA from brains obtained from infected mice and the control group (*Figure 1D*, n=5 per group). Each sample was hashed individually, samples from one time point were pooled together and sequenced. After sequencing, samples were demultiplexed for the pooled samples at the different time points. We detected a total of 44,201 nuclei (henceforth referred to as cells). Subsequently, we filtered for viable cells and removed duplicates as well as cells with high contamination of ambient mitochondrial RNA (mtRNA), which left us with a total 30,452 cells for downstream analyses. We separated the cells into neuronal and non-neuronal (NonN) cells by using known marker genes expressed in neurons (*Snap25*, *Syp*, *Syt1*, *Elavl2*; *Figure 1F and H*). As expected, NonN cells expressed in general only half the

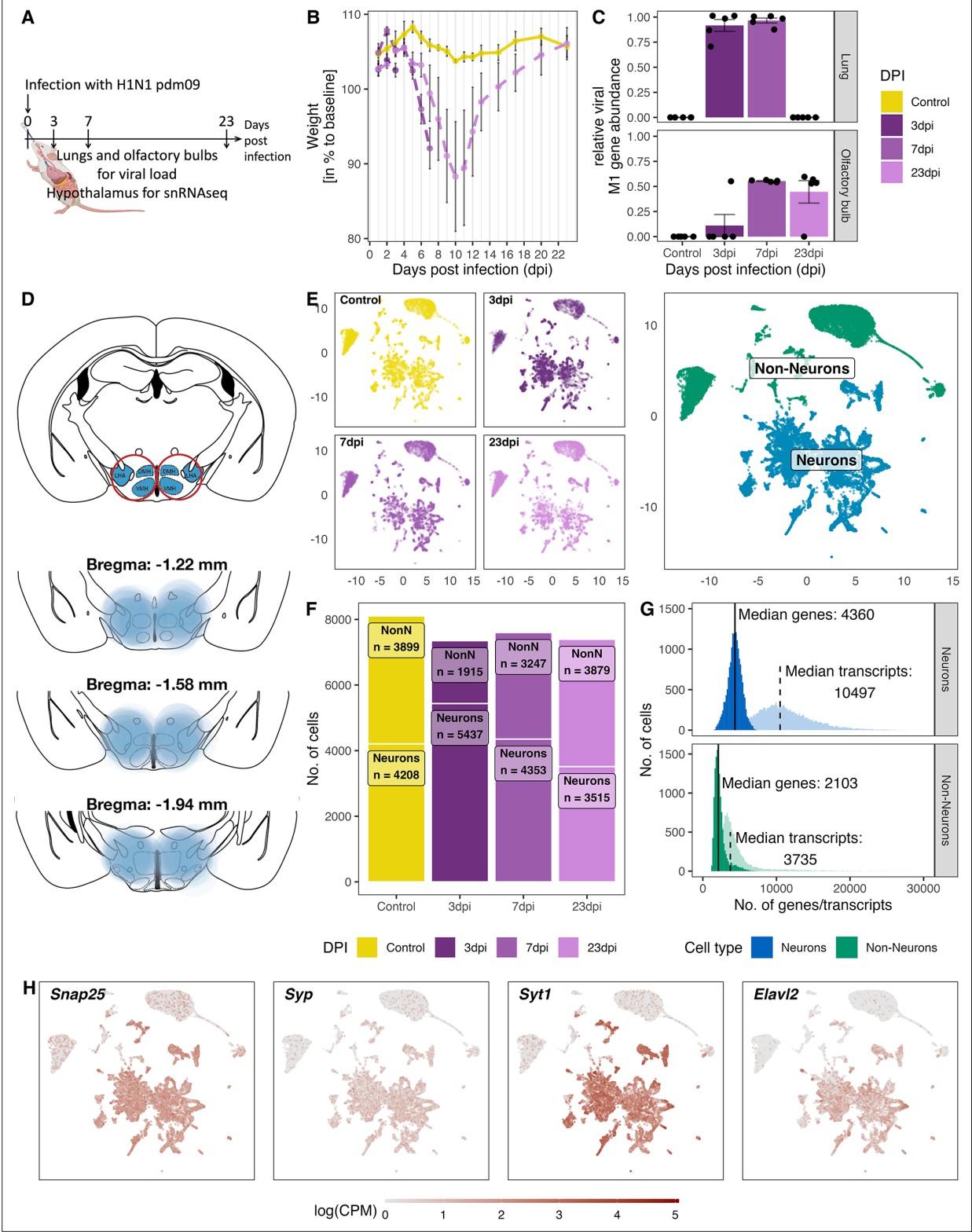

**Figure 1.** Overview of the experiment, development of the infection, and the resulting snRNA-seq dataset. (**A**) Schematic representation of the experiment. (**B**) Loss of body weight of mice during disease progression until full recovery (n=5 per group). (**C**) Viral M1 gene abundance in lung and olfactory bulb tissue at the three time points during disease progression and in controls. (**D**) Location of microdissection punches of the hypothalamus mapped to the mouse brain atlas at distances from bregma of −1.22, −1.58 and −1.94. (**E**) UMAP dimensional reduction of 30.452 cells, color-coded based on their sample group (different time points and control) membership (left) or their neuronal and non-neuronal identity (right). (**F**) Bar graph

*Figure 1 continued on next page*

Figure 1 continued

showing the cell counts of neuronal and non-neuronal cells in the different sampling groups. (**G**) Histograms depicting the distribution of transcript (lighter shading) and genes (solid shading) in all non-neuronal and neuronal cells. (**H**) Normalized expression of neuronal marker genes (*Snap25, Syp, Syt1, Elval2*) in all cells shown on a UMAP plot.

The online version of this article includes the following figure supplement(s) for figure 1:

**Figure supplement 1.** Quality control plots for the generated snRNA-seq dataset.

**Figure supplement 2.** Weight curves depicting the weight-loss due to H1N1 pdm09 infection in the different individuals.

**Figure supplement 3.** Punching location.

**Figure supplement 4.** UMAP embeddings per hash-tagged sample.

number of genes (median unique gene count: 2103) compared to neuronal cells (median unique gene count: 4360). Similarly, the median transcript count per cell in the NonN population was lower (3735 transcripts/cell) compared to the median number transcripts in neuron (10,497 transcripts/cell, 4360 features/cell; *Figure 1F*). The ratio between NonN and neuronal cells was similar at all time points except 3 dpi, where we detected a greater number of neuronal cells than NonN cells (*Figure 1G*). In addition, we performed an case-control analysis developed for comparative scRNA-seq studies (*Petukhov et al., 2022*). Cluster based analysis revealed a shift in cell densities in non-neuronal populations only in the 3 dpi cluster (*Figure 6—figure supplement 1*), possibly caused by the difference in captured nuclei at 3 dpi (*Figure 1F*). The cause of the difference in cell numbers stays unclear, however could be attributed to inconsistencies in the punching itself (*Figure 1—figure supplement 3*) and the amount of tissue used for the nuclei isolation.

## Identified NonN cell types

Among the non-neuronal cell types, we were able to identify 9 distinct populations, discriminated by their sets of cell-type-specific markers (*Figure 2A and B*). We used published datasets of different regions of the hypothalamus to confirm the identity of the distinct cell clusters (*Campbell et al., 2017*; *Chen et al., 2017*; *Mickelsen et al., 2019*; *Mickelsen et al., 2020*; *Romanov et al., 2017*). Two of the clusters belonged to the oligodendrocyte (OL) lineage, including oligodendrocyte precursor cells (OPCs, NonN_3) and mature oligodendrocytes (MOLs, NonN_1). The main marker genes in these two clusters were *Lhfpl3* and *Slc24a2* but they also expressed other typical marker genes as *Pdgfra*, *C1ql1*, *Fyn* (OPCs) or *Olig1*, *Anln*, *Plp1* in MOLs. Further, we resolved a cluster of astrocytes (NonN_2), which was defined by the differential expression of *Slc4a4* and which was also positive for *Sox9*, *Gja1* and *Agt*. The nearby cluster NonN_7 expressed similar characteristics as the astrocytes and exhibited a high expression of *Slit2*, *Col23a1*, and *Slc16a2* which have been described to be enriched in tanycytes (*Chen et al., 2017*; *Figure 2B and D*, and *Supplementary files 2 and 4*, *Figure 2—figure supplement 1*). The identity of this cluster was confirmed by label transfer of other snRNA/scRNA-seq datasets from the hypothalamus, which showed a high prediction score of this cluster as tanycytes (*Supplementary files 2 and 3*, *Figure 2—figure supplement 2*). The distinct clusters NonN_5 (*Ranbp3l+*, *Cped1+*, *Slc6a20a+*) and NonN_9 (*Bnc2+*, *Slc47a1+*, *Slc26a7+*) were identified as vascular and leptomeningeal cells, respectively. The NonN_8 cluster expressing Slco1a4 is expressing additionally *Ebf1*, *Flt1*, *Rgs5*, which are described as pericyte marker genes (*Pagani et al., 2021*; *Zeisel et al., 2018*). The cluster NonN_6 is a microglia cluster, with expression of *Hexb*, *C1qa*, and *C1qc* - known microglia marker genes. Best predictions scores for cell type label transfer were found for non-neuronal cells, with mean prediction scores 0.8 comparing the HypoMap C7 cell type levels (*Figure 2—figure supplement 1*; *Steuernagel et al., 2022*). Lastly, even though the cluster NonN_4 (*Syt1+*) is expressing general neuron markers, we could not identify any discriminatory markers for inhibitory or excitatory neuron populations (*Figure 2D*). Discriminatory marker genes such as *Slc17a7*, *Crym*, *Ptk2b*, *Itcam5*, *Slit3*, *Crym* indicate a progenitor/neuroblast identity (*Kostin et al., 2021*; *Zeisel et al., 2018*). Aligning this dataset with a dataset from the developing dentate gyrus including cells from the neuroblast lineage, indicates the presence of an immature granule subcluster strengthening the idea of NonN_4 as a neuroblast cluster (*Figure 2—figure supplement 2*; *Hochgerner et al., 2018*). However, because of the general low confidence and weak alignment we have kept this cluster unassigned.

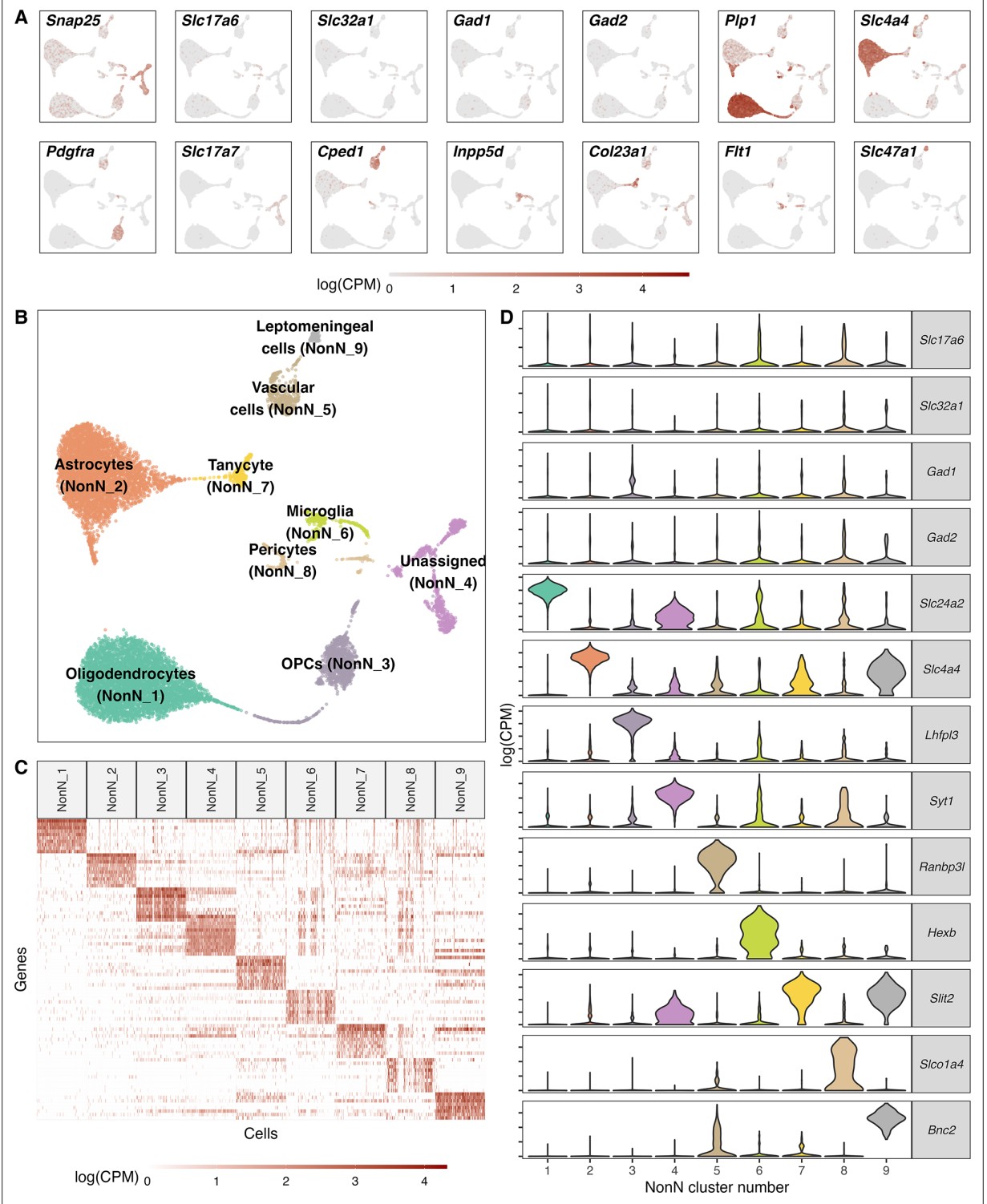

**Figure 2.** Classification of non-neuronal cell types. (**A**) Normalized expression values of different neuronal and non-neuronal cell type markers (pan-neuronal marker - *Snap25*; Glutamatergic marker - *Slc17a6*; GABAergic markers - *Slc32a1*, *Gad1*, *Gad2*; Oligodendrocyte marker – *Plp1*; Astrocyte markers – *Slc4a4*; Oligodendrocyte precursor marker – *Pdgfra*; *Cped1* – Vascular and leptomeningeal cell markers – *Cped1*, *Slc47a1*, Microglia marker – *Inpp5d*, Tanycytic marker – *Col23a1*, Pericyte marker – *Flt1*) in all non-neuronal cells (n=12.940) mapped on a UMAP. (**B**) Unsupervised clustering of non-neuronal cell types shown in a UMAP embedding and color-coded and annotated by potential cell type annotations. (**C**) Heatmap of normalized expression values showing discriminatory cell type markers of 9 non-neuronal cell populations. (**D**) Violin plots showing distribution of normalized expression values of neurotransmitters and best discriminatory cell type markers.

*Figure 2 continued on next page*

*Figure 2 continued*

The online version of this article includes the following figure supplement(s) for figure 2:

**Figure supplement 1.** Label transfer of cell-type labels from the HypoMap (*Steuernagel et al., 2022*) annotations (C7) to the Non-neuronal cell cluster here identified.

**Figure supplement 2.** Cell-type label transfer for non-neuronal cells.

## Heterogeneous populations of inhibitory and excitatory neurons

To subcluster the neurons, we first examined the expression signature of neuronal markers: Vesicular glutamate transporter 2 (VGLUT2; encoded by *Slc17a6*); Vesicular GABA Transporter (VGAT, encoded by *Slc32a1*), glutamate decarboxylase 67 (GAD67; encoded by *Gad1*); and histidine decarboxylase (encoded by *Hdc*) and separated the neuronal cells in GABAergic and glutamatergic cells (*Figure 3A*). We identified a total of 26 GABAergic and 30 glutamatergic clusters. Within the glutamatergic cells we found also a small population of histaminergic (Glut_24; *Hdc*+) and dopaminergic neurons (Glut_27; *Th*+, *Ddc*+). The histaminergic neurons were left for visualisation and analysis purposes in this cell type, despite them often being described as GABAergic (*Figure 3B*). Among the glutamatergic cells, we were able to detect two small clusters of HCRT expressing neurons (Glut_29; *Hcrt*+) and MCH expressing neurons (Glut_25; *Pmch*+). The cellular resolution aligns well with the composition of other cell types seen in other studies of the hypothalamus (*Campbell et al., 2017*; *Chen et al., 2017*; *Mickelsen et al., 2019*; *Mickelsen et al., 2020*; *Romanov et al., 2017*; *Steuernagel et al., 2022*). Among well-described neuronal cell cluster, we found a good overlap with *Hcrt*+, *Pmch*+, *Hdc* +and Prkch +neurons (*Figure 3—figure supplement 1*) and *Pomc*+, *Nkx2−4*+GABAergic neurons (*Supplementary file 7*). Additionally, we identified a *Argp*+/*Npy* +co-expressing cell cluster as a sub-cluster of GABA_1 (*Figure 8—figure supplement 2*).

## General transcriptional disease responses within the medial hypothalamus across all cells

To understand the nature of the molecular changes between the control group and the infected groups at different time points of disease progression, we first used a pseudo-bulk approach to compare their transcriptional signatures of all cells per time point against each other. A principal component analysis based on the pseudo-bulk dataset showed a clear clustering of the biological replicates per time point (*Figure 4A*). We found a clear separation of the samples taken at 3 dpi from the other time points. Seven dpi is clustering separately between 3 dpi and the control samples, whereas the samples from the recovery group at 23 dpi are spread between the 7 dpi and the control group cluster. This indicates a clear transcriptional change during the first 7 days of infection. The recovery group is mostly overlapping with the control group, but one sample clusters together with the 7 dpi group, suggesting a slower disease or recovery process in this animal (Animal 23.4, *Figure 4A*). The overall gene expression counts show a pattern of lower counts per cell at 3 dpi in both GABAergic and glutamatergic cell populations. In glutamatergic cells this is a lasting but less pronounced effect until 23 dpi (*Figure 4B*, *Supplementary file 5*). Next, we performed a differential gene expression (DGE) analysis comparing the control samples to samples from different time points during disease progression, disregarding any differentiation between cell subpopulations. We tested, based on previous studies, gene sets know to be involved in neuroinflammatory and immune regulatory process within the CNS for possible changes in gene expression (*Dusedau et al., 2021*; *Hosseini et al., 2018*; *Sadasivan et al., 2015*; *Zielinski et al., 2013*). We primarily detected a down-regulation of these gene categories, which was strongest at 3 dpi with only very few significant changes lasting until 23 dpi (*Figure 4C*).

The only up-regulated genes were detected at 3 and 7 dpi belonging to the category of neuron-related factors. Genes in the other categories were consistently down-regulated or not significantly regulated at all. This also included the virus-related cytokines and IFN-response genes (i.e. *Il1a*, *Ifih1*, *Figure 4C*) at 3 dpi. Toll-like receptors (TLRs) are central in initiating the innate immune response by orchestrating signalling pathways leading to the release of inflammatory cytokines (*Kawai and Akira, 2011*). In our dataset, we see a general downregulation of TLRs (*Tlr2*, *Tlr3*, and *Tlr4*, *Figure 4C*). These results show that a peripheral H1N1 pdm09 infection cause decreased gene expression in non-neuronal, GABAergic, and glutamatergic cell populations in the hypothalamus affecting several immune related processes.

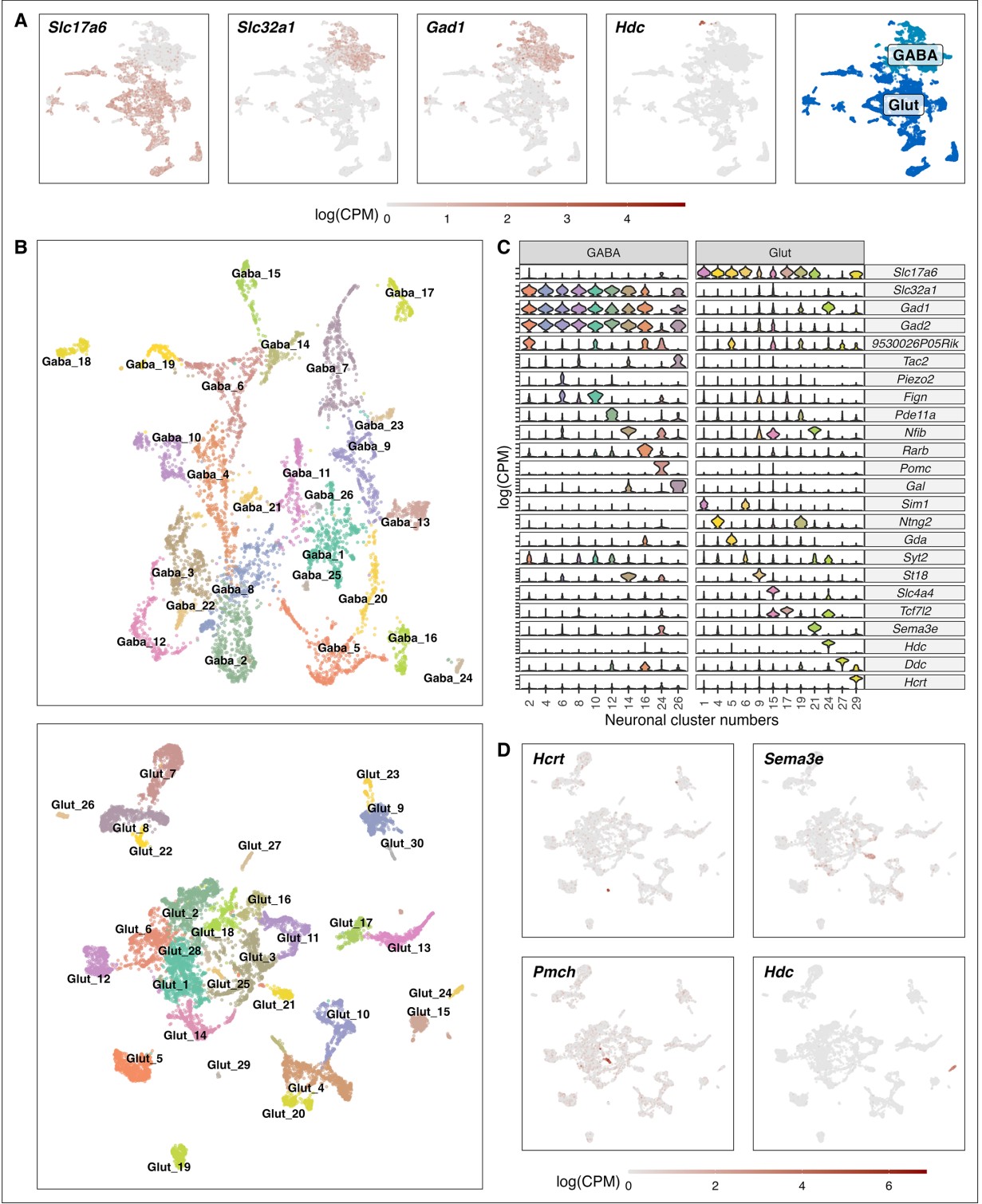

**Figure 3.** Classification of GABAergic and glutamatergic neuronal cell types in the hypothalamus. (**A**) Normalized expression values of different neurotransmitters (Glutamatergic marker - Slc17a6; GABAergic markers - *Slc32a1*, *Gad1*, *Gad2*) in all neuronal cells (left) and a color-coded UMAP projection based on their GABAergic or glutamatergic identity (right) (n=17.512). (**B**) Un-supervised clustering of GABAergic (upper, n=6.032) and glutamatergic (lower, n=11.481) cells in UMAP plots. Cell type clusters are color-coded and annotated with labels. (**C**) Violin plots showing normalized expression values of neurotransmitters and discriminating marker genes of selected GABAergic (left) and glutamatergic (right) cell type clusters. (**D**) Glutamatergic UMAP plots showing normalized expression values of distinct markers for hypothalamic neuron populations (HCRT neurons – upper left, PMCH neurons – lower left, GnRH neurons – upper right, Histaminergic neurons – lower right).

*Figure 3 continued on next page*

*Figure 3 continued*

The online version of this article includes the following figure supplement(s) for figure 3:

**Figure supplement 1.** Label transfer of cell-type labels from the HypoMap (*Steuernagel et al., 2022*) annotations (C285) to glutamatergic cell cluster here identified.

**Figure supplement 2.** Label transfer of cell-type labels from the HypoMap (*Steuernagel et al., 2022*) annotations (C285) to GABAergic cell cluster here identified.

**Figure supplement 3.** Cell-type label transfer for glutamatergic cells.

**Figure supplement 4.** Cell-type label transfer for GABAergic cells.

## Specific molecular regulation in non-neuronal, glutamatergic, and GABAergic cell populations

Next, we resolved the molecular disease responses in better detail, by performing a differential gene expression (DGE) analysis on the three overall cell populations (GABAergic, glutamatergic, and NonN). Similarly, to the previous pseud-bulk approach, we detected the strong down-regulation of genes at 3 dpi in all the three distinct cell populations (*Figure 5A*). The overall number of down-regulated genes decreased at 7 and 23 dpi, but more differential expressed genes were detected in the GABAergic cells at 23 dpi, than at 7 dpi (*Figure 5A and B*). Of all highly regulated genes at 3 dpi (False discovery rate (FDR) ≤ 0.05, log fold change () >1 or logFC < –1), approximately a quarter (27%) of the genes were shared between the 3 cell populations, and 30% were shared between the two neuronal clusters. Around 10% of the genes expressed in the GABAergic and glutamatergic cell clusters at 3 dpi were found expressed only in these cells (*Figure 5B*). In contrast, 20% of the DEGs in the NonN cells at 3 dpi were unique to these cells only. At 7 dpi, only 4 genes were shared between all 3 clusters, and 59 genes between the two neuronal clusters (*Figure 5B*). At 23 dpi no more DEGs were detected in the NonN cells. Most differentially expressed genes (DEGs) at this time point belonged to the GABAergic cluster. To exclude if persistent transcriptional effects reported in neurons but not in non-neurons are caused by the different amount of captured transcripts between neurons and non-neurons (*Figure 1G*), we created a down-sampled dataset with similar transcripts per cell in neurons and non-neurons (*Figure 5—figure supplement 1B*). Here we reported the same effect, of high down-regulation of genes in all cells (*Figure 5—figure supplement 1A*) at 3 dpi, with a less pronounced pattern at 7 dpi. In the recovery group at 23 dpi, we found again a transcriptional response only in the neuronal cells but nit in non-neuronal cells (*Figure 5—figure supplement 1A*). To study what functional effects these detected changes in gene expression might have, we performed a gene ontology enrichment analysis. *Figure 5C* shows all the significantly regulated pathways. Of special interest, we detect an enrichment of genes involved in immune responses at 3 dpi in all three clusters, while this is restricted to the neuronal clusters at 7 dpi and only the Glut cluster at 23 dpi (*Figure 5C*). Further, enrichment of neuropeptide signalling, and hormone activity pathways are also present in all three clusters at 3 dpi, while lasting until 23 dpi only in the neuronal cells. These data suggest that long term effects in the hypothalamus after a peripheral IAV infection affects primarily neurons.

## Dissecting immune system processes in microglia

We tested all identified NonN and neuronal cell populations for differential gene expression. A cell cluster was excluded from analysis if their cells were originating from less than three of the biological replicates and if they were composed of less than two cells per cluster per biological replicate. Similarly, we used for further downstream analysis only strongly expressed genes with high up- or down-regulation (FDR ≤ 0.05, logFC >1 or logFC<-1). As shown before, the strongest regulatory effect on the different clusters' transcriptomes were detected at 3 dpi, which lasted partially until 23 dpi (*Figure 6A*). Several GABAergic clusters down-regulated more genes at 23 dpi than at 7 dpi. These cell populations (GABA 2, 4, 6, 8, 10, 12) are aligning close to each other in the dimensional reduction (*Figure 3B*) and are overlapping in the centre of the UMAP. The glutamatergic neuron populations show in general a similar pattern, with the strongest transcriptional response at 3 dpi and less differentially regulated genes at 7 and 23 dpi.

As described in the previous paragraph, we identified immune response pathways as one of the most enriched gene ontologies (GO) terms in the dataset. Therefore, we selected only the genes

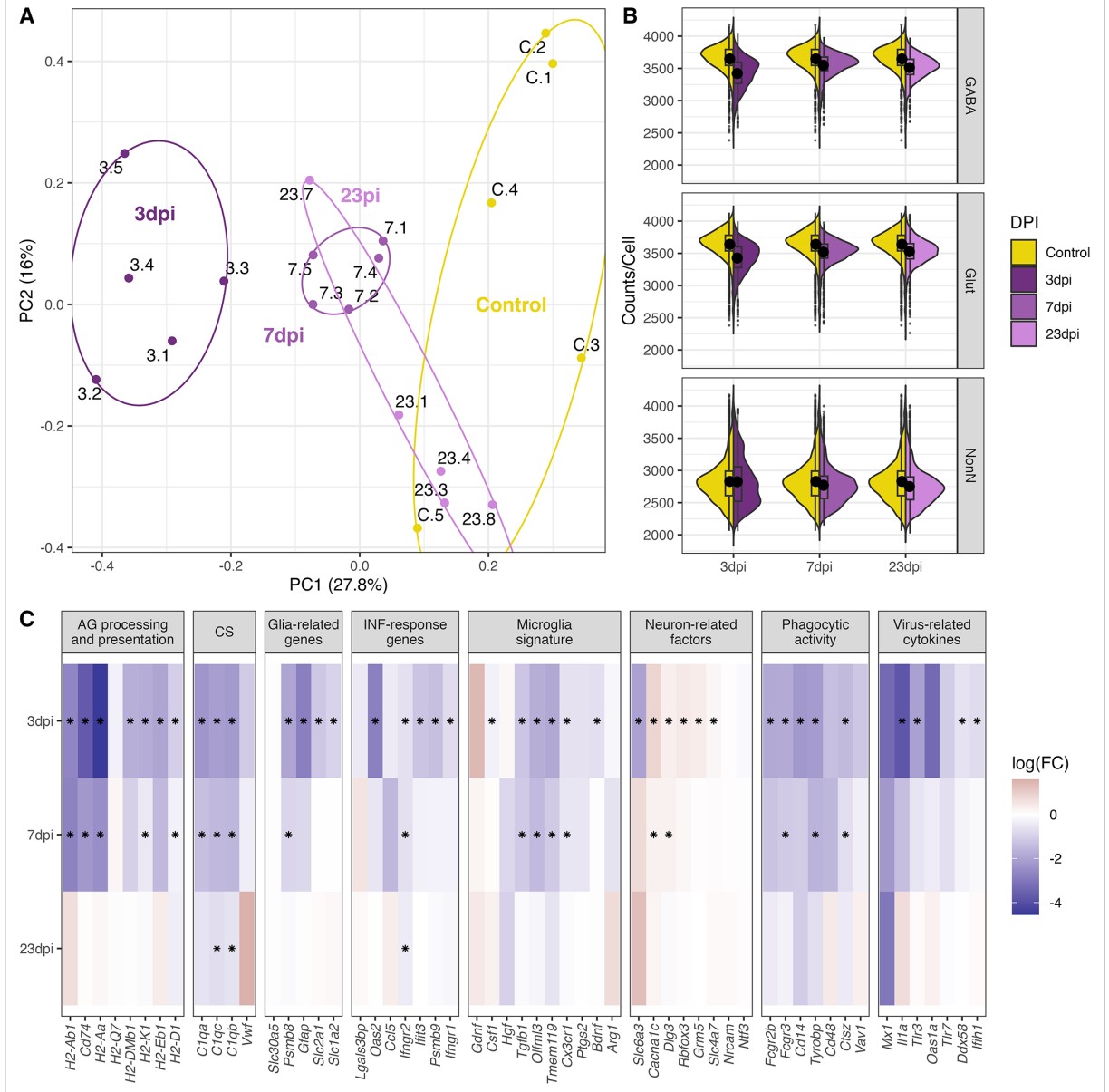

**Figure 4.** General changes in the transcriptomic landscape of the hypothalamus during peripheral IAV infections. (**A**) Principal component plot of combined counts per sample. Each sample snRNA-library were down-sampled to a total of 300 cells (100 non-neuronal, 100 GABAergic, 100 glutamatergic cells) before normalized counts were aggregated before principal component analysis. Samples are color-coded according to their group membership and the numbers correspond to the individual animal IDs. (**B**) Violin plot showing the distribution of counts per cell at the different time points. The three points of infections are compared to the control group. (**C**) Heatmap showing log-transformed log-fold changes of known immune-related genes. Black stars indicate significant expression (FDR ≤ 0.05).

annotated in these immune-related GO terms, leaving us with 22 genes. Interestingly, all these 22 immune related genes are regulated in microglia (NonN_6) at 3 and 7 dpi (*Figure 6C*). In other non-neuronal clusters less than half of this genes are differentially regulated at 3 dpi and none at 7 or 23 dpi. Similarly, neuronal cell populations showed fewer genes regulated at 3 dpi than microglia, however especially some GABAergic cell types show a transcriptional response of immune genes until 7 and even 23 dpi (i.e. GABA_7, 16 and 17). The here reported highest regulation of immune genes indicate that microglia are also in the medial hypothalamus the main immune orchestrating cell population.

We next aimed to describe the transcriptional changes in the microglia during disease progression. Raw DEG numbers emphasize the microglia cluster as the only cell cluster with a strong upregulation

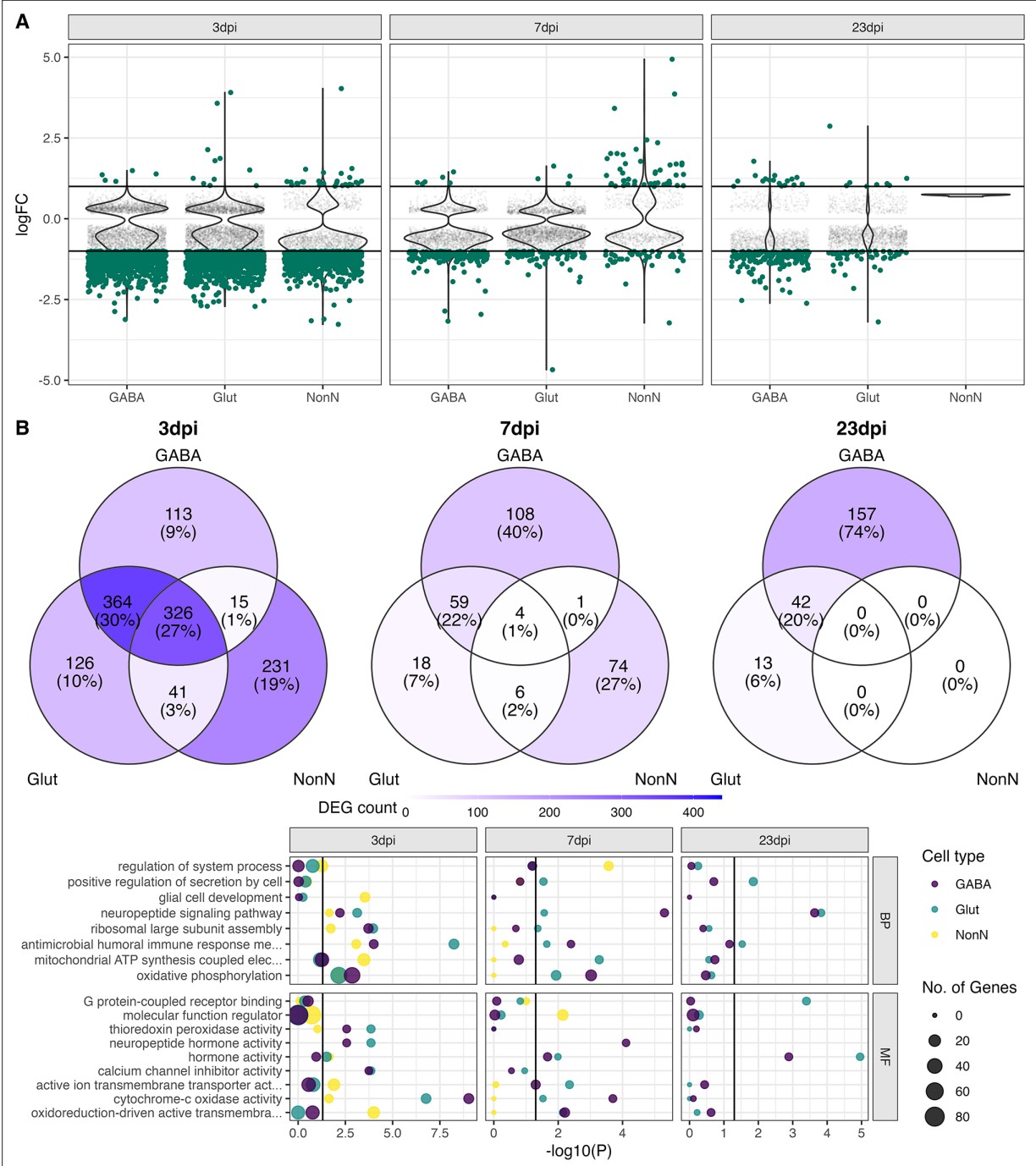

**Figure 5.** Transcriptional changes in GABAergic and glutamatergic neurons of the hypothalamus during peripheral IAV infections. (**A.**) Violin plots showing the log-transformed fold changes (logFC) per gene calculated for the GABAergic, glutamatergic, and non-neuronal cells at 3, 7, and 23 dpi. Dark-green dots represent significantly differential expressed genes (FDR ≤ 0.05) with a logFC greater than 1 or lower than –1. Small grey dots show significantly expressed genes with a logFC between –1 and 1. (**B**) Venn diagrams depicting the overlap of highly differential expressed genes (logFC >1 or logFC < –1) between the three main cell type clusters at each time point. (**C**) Gene ontology analysis showing the most significant ontology terms for the category of Biological Process (BP, top) and Molecular Function (MF, bottom). The two highest significant ontology's for each time point and each cell type were chosen if it included more than 4 annotated genes. The size of the dots indicates the number of genes included in the ontology term and the colour show the cell type annotation.

The online version of this article includes the following figure supplement(s) for figure 5:

**Figure supplement 1.** DEG analysis in down-sampled dataset.

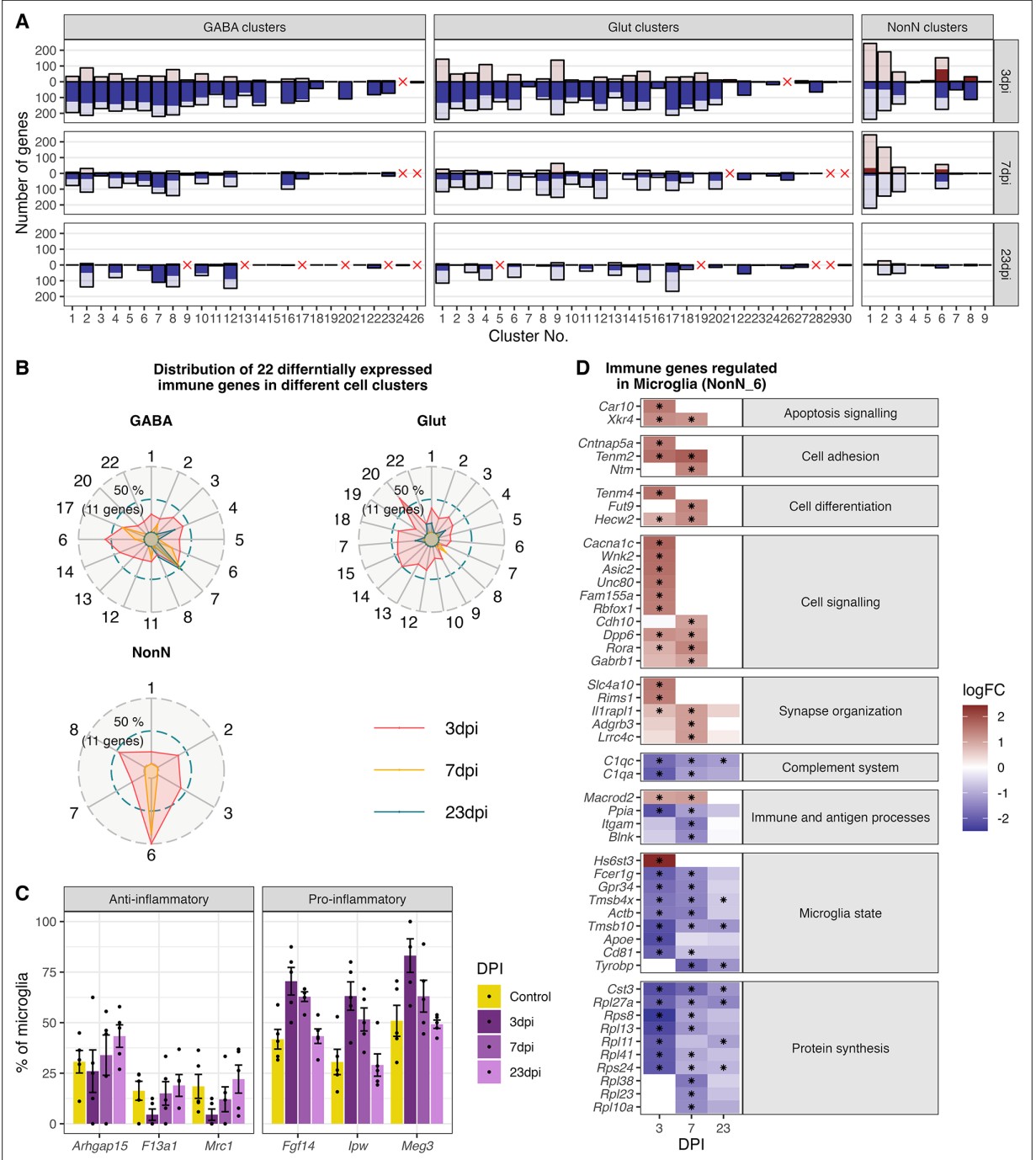

**Figure 6.** Gene expression changes in different cell type clusters and microglia activation during peripheral IAV infections. (**A**) Bar plot showing the number of significantly differentially expressed genes in each cell type cluster at 3, 7, and 23 dpi. Dark solid blue and red bars show the number of highly differential expressed gene (FDR ≤ 0.05, logFC <= –1 or logFC ≥ 1) per cell type cluster and time point. Lighter shaded bars depict the number of all significant regulated genes (FDR ≤ 0.05, logFC ≥ 0 or logFC ≤ 0). Differentially expressed genes (DEGs) were only included for clusters containing at least three cells per sample and per time point in the cluster. A red cross indicates instances where these criteria were not met. (**B**) Radar plot showing the number of significantly and strongly regulated genes (FDR ≤ 0.05, logFC ≥ 1 or logFC<= –1: n=22) involved in immune processes (based on their gene ontology annotations) per cell cluster at the different time points (red: 3 dpi, yellow: 7 dpi, blue: 23 dpi). (**C**) Percentage of the number of cells expressing distinct anti- or pro-inflammatory signature genes in the microglia cluster. Cells with a raw count of 0 for a gene were assumed as non-expressing, all other cells were assumed to express this gene. Shown are percentages of microglia cells expressing a gene compared to all microglia at each time point across all samples. (**D**) Relative changes in expression levels of disease associated genes within the microglia cluster. Data is show as

*Figure 6 continued on next page*

*Figure 6 continued*

log-transformed fold changes (logFC) of each time point compared with control group. Black stars depict significant differentially expressed genes (FDR ≤ 0.05).

The online version of this article includes the following figure supplement(s) for figure 6:

**Figure supplement 1.** Cluster-based composition shifts calculated by Cacoa.

**Figure supplement 2.** Cluster-based analysis of changes in expression magnitudes.

**Figure supplement 3.** Cluster-free expression shifts.

**Figure supplement 4.** Identified gene programs at 3 dpi based on cluster-free genes expression analysis.

**Figure supplement 5.** Identified gene programs at 3 dpi based on cluster-free genes expression analysis.

**Figure supplement 6.** Identified gene programs at 3 dpi based on cluster-free genes expression analysis.

of genes at 3 dpi (75 genes), and a long-lasting down-regulation of genes until 23 dpi (52 and 12 genes at 7 and 23 dpi respectively; *Figure 6A*, cluster NonN_6), indicating the importance of microglia for immune homeostasis in the medial hypothalamus. A closer look at the highest up-regulated genes reveals a strong induction of genes involved in cell adhesion, signalling and differentiation, synapse organization as well as apoptosis signalling. Down-regulated genes participate in the complement system, immune and antigen processing, microglia state genes and ribosomal genes contributing to protein synthesis (*Figure 6E*).

Typical markers like cell surface proteins or intracellular markers are often used to identify pro- (M1 state; i.e. *Il1b*, *Ifng*, *Tfna*, *Fcgr3*) or anti-inflammatory (M2 state; i.e. *Tgfb*, *Mrc1*, or *Arg1*) states of microglia (*Jurga et al., 2020*; *Moehle and West, 2015*). These markers were not regulated during disease progression in our dataset. Discrimination of the activation state of microglia using snRNA-seq has been reported to be challenging, due to a depletion of certain marker genes in the nucleus especially in human tissue (*Thrupp et al., 2020*). Because of this lack of typical M1 and M2 markers, we identified alternative DEGs in microglia at different the time points related to activation states. Two long non-coding RNA (lncRNA) genes: Maternally Expressed 3 (*Meg3*), Imprinted in Prader-Willi Syndrome (*Ipw*) and the gene Fibroblast Growth Factor 14 (*Fgf14*) which have been recently connected to immune activation and pro-inflammatory microglia states (*Liu et al., 2021*; *Meng et al., 2021*; *Senatorov et al., 2019*), were found in our dataset enriched in the microglia at 3 and partly at 7 dpi (*Figure 6D*). In contrast, genes connected to an anti-inflammatory state (*F13a1*, *Mrc1*, *Arhgap14*), were expressed in less microglia cells during acute infection than in control and 23 dpi. The results show that microglia are also in the medial hypothalamus a main immune orchestrating cell population, shifting from their homeostatic state towards a pro-inflammatory microglia population during 3 and 7 dpi. This shift is completely resolved after recovery at 23 dpi.

## Identification of oligodendrocyte and astrocytes subclusters at 7 dpi

In oligodendrocytes (NonN_1) and astrocytes (NonN_2) between 144–244 genes were up- or down-regulated (logFC >0 or logFC <0) at 3 and/or 7 dpi (*Figure 6A*). This was accompanied by a shift of the cells in the dimensional reduction plot of the not integrated data, especially at 7 dpi (*Figure 7A*). In the oligodendrocytes this was accompanied by an up-regulation of genes involved in stress-related and glucocorticoid pathways (*Figure 7B and C*). The same was not seen in the astrocyte sub-cluster. We identified significantly regulated GO terms associated with transport pathways in oligodendrocytes, with a strong up-regulation of genes in these terms at 3 and 7 dpi (*Figure 7D*). Only mitochondrial transport processes and transmembrane transport included down-regulated genes. Genes associated with GABAergic transport was up-regulated in oligodendrocytes uniquely at 7 dpi. Marker genes for the astrocytic 7-dpi cluster were identified as *Aldh1a1*, *Lrrc8a* (annotated here as ENSMUSG00000099041) and *Zfhx3*. *Zfhx3* showed a very distinct expression in this cluster (*Figure 7C*). For astrocytes, the GO term oxidoreductase activity was highly and uniquely enriched at 7 dpi (*Supplementary file 8*). Accordingly, we found the oxidoreductase genes to be highly up-regulated in the 7 dpi cluster, with only a few being also regulated a 3 dpi and non at 23 dpi (*Figure 7E*). Together these results suggest that an IAV infection induces slowly evolving transcriptional changes in oligodendrocytes and astrocytes, leading to the appearance of distinct sub-populations of these cell types at 7 dpi. This altered cell stated disappear again after recovery at 23 dpi.

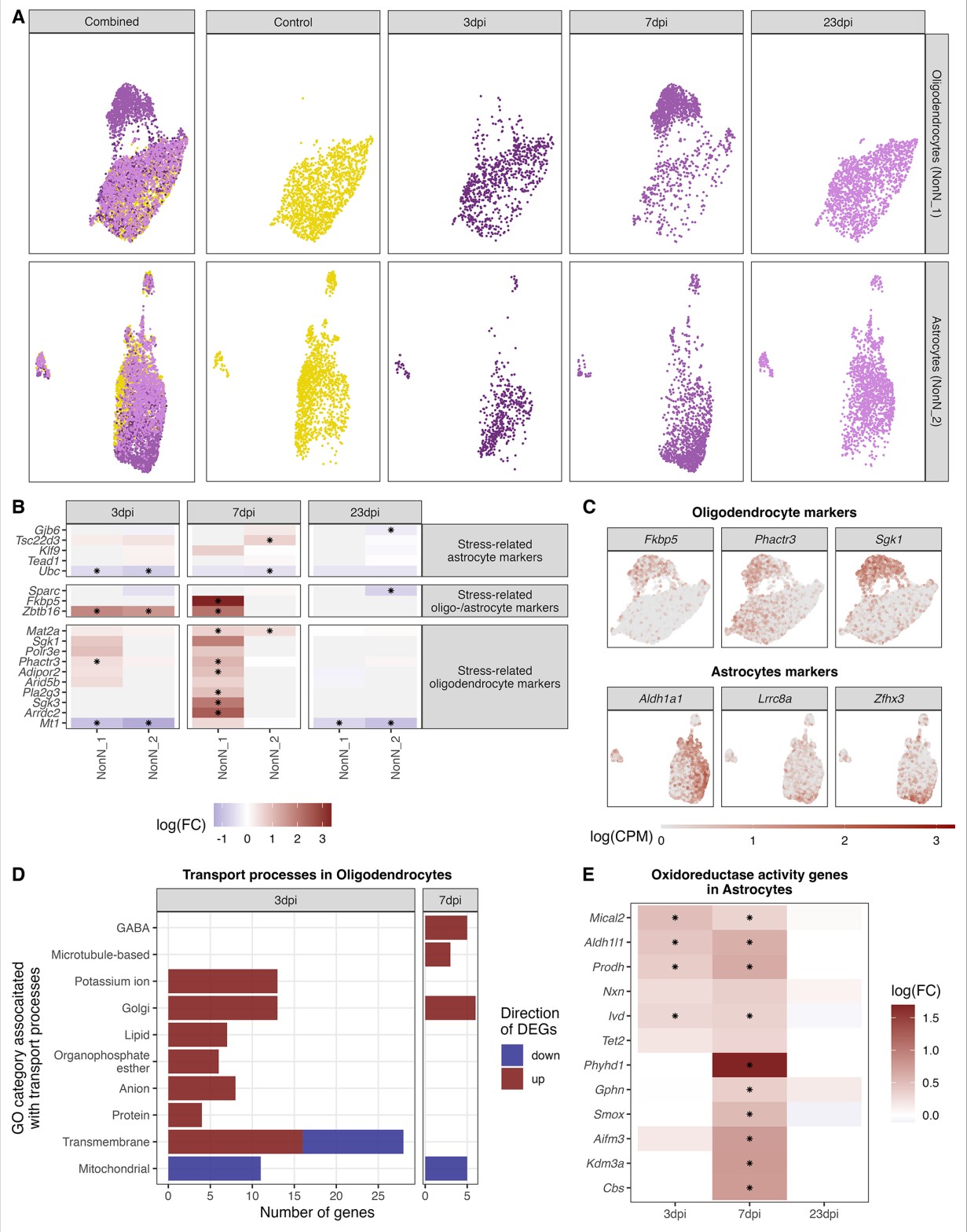

**Figure 7.** Identification of distinct oligodendrocyte and astrocyte sub-clusters at 7 dpi. (**A**) Combined UMAP plot of all cells of the oligodendrocyte (upper right, n=5.023) and astrocyte (lower right, n=4.542) cluster in controls and at different time points. (**B**) Relative gene expression changes (in comparison of to the mock-infected group) of stress-related oligodendrocyte and astrocyte markers within the oligodendrocyte (NN_1) and astrocyte (NN_2) subcluster. Data is shown as log-transformed fold changes, an asterisks indicates statistical significance (FDR ≤ 0.05). (**C**) Normalized expression values of selected marker genes at 7-dpi in UMAP plots of all oligodendrocytes or astrocytes. (**D**) Number of differentially expressed genes (FDR ≤

*Figure 7 continued on next page*

*Figure 7 continued*

0.05, logFC <= –1 or logFC ≥ 1) annotated to gene ontologies associated with transport process in the oligodendrocytes at 3 and 7 dpi. (**E**) Expression dynamics of differential expressed oxidoreductase genes in the astrocyte cluster at 3, 7, and 23 dpi. A black star indicated significant differential expression (FDR ≤ 0.05).

The online version of this article includes the following figure supplement(s) for figure 7:

**Figure supplement 1.** Expression of different GABAergic and glutamatergic transporters.

## Long-term transcriptional response in neurons following as IAV infection

Even though the most prominent feature of the hypothalamic transcriptional response to H1N1 pdm09 was the downregulation of genes at 3 dpi across all cell clusters, some neuronal cell populations showed long-term differential expression of genes. As described in previous paragraphs, we detected overall changes in both GABAergic and glutamatergic neurons at 23 dpi (*Figure 5B*). When looking at neuronal subclusters, we found that 6 out of 26 GABAergic cell clusters and 17 out of 30 glutamatergic cell clusters contained significantly regulated genes at 23 dpi when compared to the control group (*Figure 6A*). The cluster based case-control analysis (Cacoa) showed a similar dysregulation of neuronal and non-neuronal cell populations at 3 and 7 dpi (*Figure 6—figure supplement 2*). However, the long-lasting transcriptional changes described before, were not reproduced with this cluster-based analysis. Though, analysis of cluster-free expression shifts shows dysregulation of neuronal and non-neuronal cells at the acute phase of the infection (*Figure 6—figure supplements 4–6*), however detects mainly transcriptional changes in neurons at 23 dpi (*Figure 6—figure supplements 3 and 6*). Our initial GO term enrichment analysis (*Figure 5C*) showed several different pathways being affected at all timepoints, so we aimed at dissecting this further by analysing individual neuronal cell clusters. In this analysis we included glutamatergic and GABAergic cell clusters, with a high DGE profile and at least 10 genes exhibiting a logFC greater than 1 or less than –1. The 15 neuronal clusters fulfilling this criterion were: Gaba_2, 4, 6, 8, 10, 12 and Glut_1, 4, 6, 9, 11, 15, 17, 18 and 20. In general, we found the same regulatory processes to be regulated across many of the neuronal cell types. When looking at the pathways that remain altered at 23 dpi these include translational processes in the cytosol, metabolic processes in the mitochondria, as well as apoptotic processes (i.e. mitochondrial respiration and TCA cycle) (*Figure 8*, *Supplementary files 7 and 8*).

Our analysis shows that many of the neuronal subpopulations respond with a similar pattern when the organism is challenged by an infection. To determine whether there would also be differences in response pattern between neuronal subtypes, we selected six hypothalamic neuronal subtypes with known regulatory functions in metabolism: Agrp+/Npy+, Lepr+, and POMC+ neurons; and sleep: Hdc+, Pmch+, and Hcrt+ neurons. With both feeding and sleep patterns changing during influenza, we would expect these neuronal subtypes to change their gene expression during the experiment. This is indeed what we find. While most of the subtypes respond with overall downregulation of gene expression (*Figure 8C–D*), the POMC+ neurons show a different pattern with equal and strongly both up and downregulated genes.

Taken together, neuronal populations were the only cell populations that were affected by the H1N1 pdm09 infection until recovery, with changes including inhibition of genes contributing to metabolic and translational processes in the neurons. While most neuronal subpopulations follow this same regulatory pattern, POMC+ neurons show bigger and more unique changes in gene regulation following a peripheral influenza infection.

## Discussion

The compiled snRNA-seq dataset presented above, to our knowledge, represents the first comprehensive transcriptome profiling on single-nuclei level of the hypothalamus during a peripheral IAV infection. We detected 26 GABAergic, 30 glutamatergic and 9 non-neuronal cell types, aligning closely with previously published datasets of the hypothalamus (*Campbell et al., 2017*; *Chen et al., 2017*; *Mickelsen et al., 2019*; *Mickelsen et al., 2020*; *Romanov et al., 2017*; *Zeisel et al., 2018*). In addition, we deconvoluted the molecular changes of distinct cell populations in the hypothalamus during a peripheral IAV infection and detected a whole range of transcriptional changes across both NonN

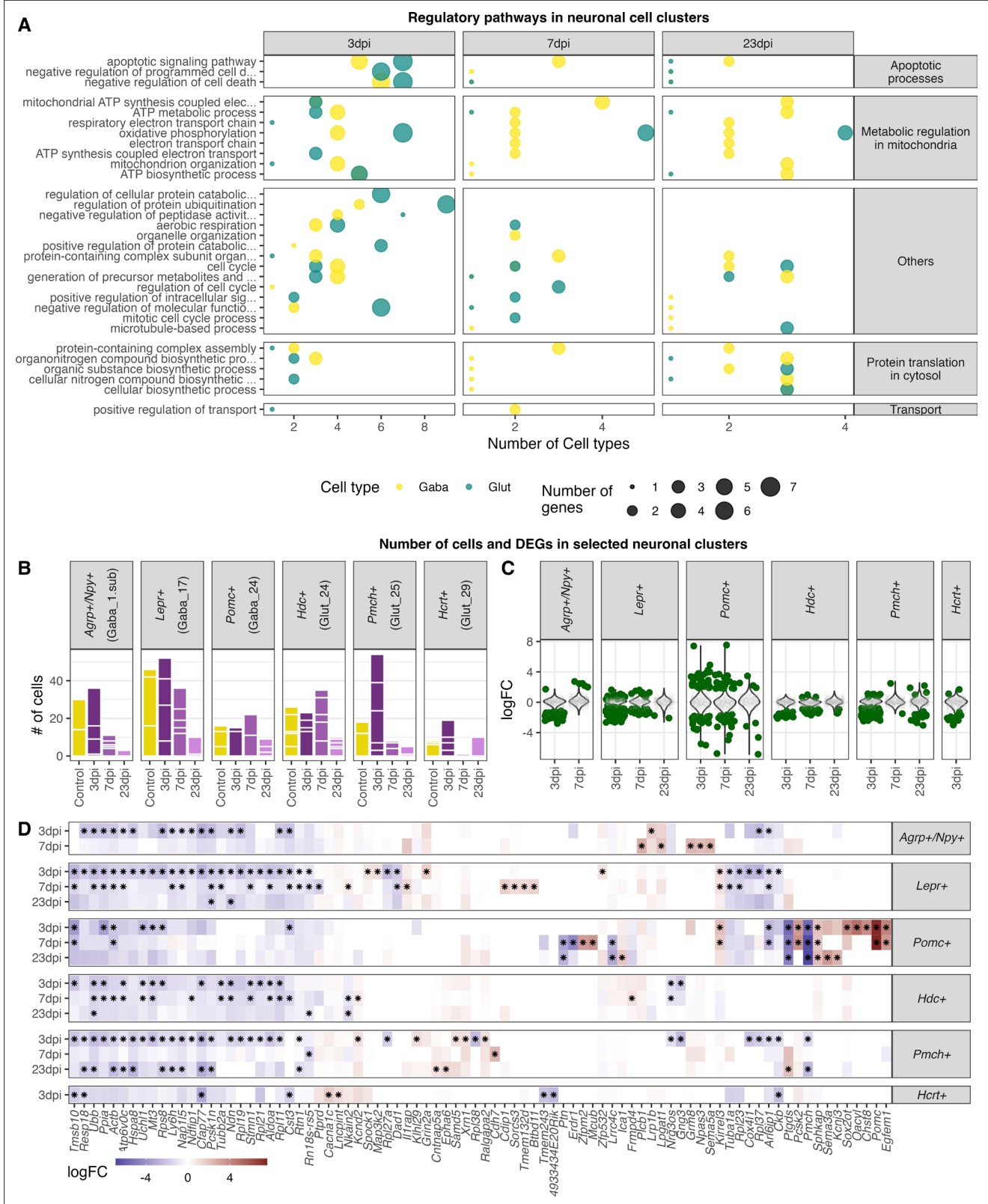

**Figure 8.** Highly enriched gene ontology categories in neuronal cell populations at different time points of infections. (**A**) Dot-plot depicting the number of neuronal cell populations significantly enriched gene ontology categories. Only cell clusters with at least 10 significantly differently expressed genes (logFC <= –1 or logFC ≥ 1, FDR≤ 1) in all three analysed time points were included in the here presented plot. Neuronal cell clusters included are (GABA_2, 4, 6, 8, 10,12; Glut_1, 3, 4, 6, 9, 11, 13, 15, 17). Only significantly enriched (*P*elim ≤ 0.05) gene ontology terms were chosen and for each

*Figure 8 continued on next page*

*Figure 8 continued*
cluster per time point the 10 most enriched GO terms were included. (**B**) Bar plot depicting the amount of captured nuclei in the individual animals at the different time points for known hypothalamic neuron populations. (**C**) Differential expressed genes in selected neuron populations. Dark green dots show significantly differential expressed genes (FDR ≤ 0.05). (**D**) Heatmap showing the five highest and lowest DEGs (FDR ≤ 0.05) in each neuronal population.

The online version of this article includes the following figure supplement(s) for figure 8:

**Figure supplement 1.** Reactome pathway enrichment of differentially expressed genes in neurons.

**Figure supplement 2.** Identification of an *Agrp*+/Npy +neuron cluster.

and neuronal cell types. In addition, we believe our dataset is a valuable resource for the scientific community to explore unsolved questions of hypothalamic responses during peripheral infections.

We were able to report viral infection of the olfactory bulb using H1N1 pdm09. This is comparable to what has been seen in previous studies using the non-neurotropic H1N1 strain PR8 (*Majde et al., 2007*; *Zielinski et al., 2013*). The same studies found replication intermediates of the viral nucleoprotein (NP1), suggesting the potential ability of the virus to replicate further within the OB. However, another mechanism has been suggested, in which the virus continuously replicate in the nasal epithelium and then gets transported into the OB (*Zielinski et al., 2013*). Other areas deeper in the CNS, like the hippocampal region, did not show any viral infiltration by PR8. (*Hosseini et al., 2018*; *Jurgens et al., 2012*). Similar results were reported for infections with H1N1 pmd2009, where the virus could not be detected in the hippocampus, striatum or cortex, supporting the hypothesis that this virus is not replicating in the CNS (*Sadasivan et al., 2015*).

In contrast to our results, studies using the PR8 disease model reported elevated mRNA expression levels of cytokines related to viral infections 24 hr and 6 days post infection (*Zielinski et al., 2013*). We found in general a strong down-regulation of genes at 3 dpi, including also virus-related cytokines and IFN-response genes (i.e., *Tlr3*, *Il1a*, *Ifih1*) at 3 dpi with no differential regulation of these transcripts at 7 dpi. This indicate that we do not have neuroinflammatory processes in the hypothalamus in our model, or that the here chosen time point of 3 dpi might be too late to capture innate immune activation in the CNS.

Even though we used different published datasets to annotate the here-in identified cell types or compare gene expression changes in a disease context, the comparability of these datasets is

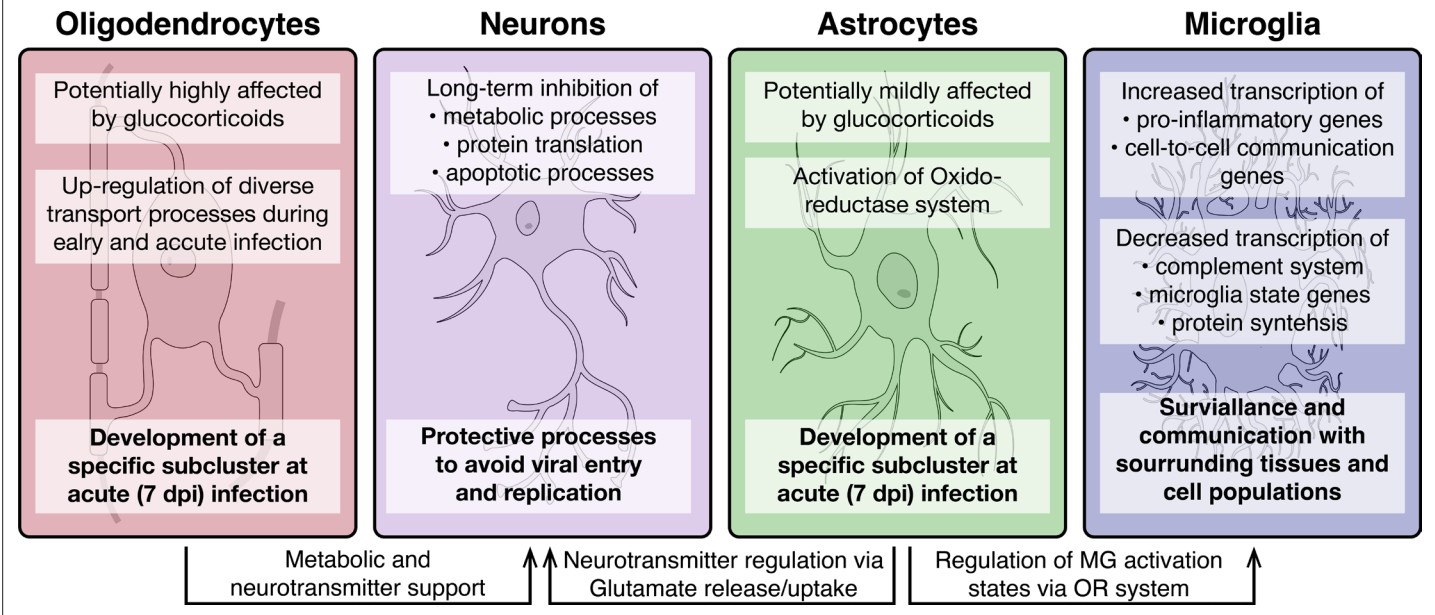

**Figure 9.** Overview of molecular processes occurring in different cell types within the hypothalamus during a peripheral H1N1 IAV infection. Schematic overview of the main molecular mechanisms of oligodendrocytes, astrocytes, microglia and neurons during acute to late immune responses of a peripheral IAV infection with the H1N1 pdm09 Influenza A virus.

challenging due to differences in the experimental setups. We identified a lack of RNA sequencing studies focusing solely on molecular changes in the hypothalamus during non-neurotropic IAV challenges. Different molecular changes based on viral strain or between brain regions are expected and further studies are needed to resolve this.

A prominent feature of our dataset is a long-lasting down-regulation of gene transcripts until 23 dpi in many neuronal cell populations. Gene set enrichment analysis showed these to be mainly related to processes in mitochondria (i.e. citric acid cycle (TCA), ATP synthesis) or to protein translation in the cytosol. This could be interpreted as a defensive mechanism, for the neurons to escape viral entry or replication in the cell and hence protect the cells from apoptosis. Different intermediates of the citric acid (TCA) cycle can be important for viral replication within the cell. In the context of IAV, acetyl-CoA and ATP are known to be beneficial for viral replication. Acetyl-CoA is also used for viral acetylation and ATP is needed for budding of the virus (*Hatakeyama et al., 2018*; *Hui and Nayak, 2001*; *Sánchez-García et al., 2021*). The downregulation of genes contributing to the translational machinery could be another attempt to inhibit translation of viral proteins in case of a potential entry of the virus into the cell. Another explanation for our findings could be that the downregulation of metabolic and translational processes in the cell might be caused by the excessive weight loss over the disease course and the lack of energy resources (*Figure 9*). However, we did see these changes also after full weight regain at 23 dpi. In addition, our results indicate that the virus is present in the OB until complete recovery at 23 dpi, suggesting this as one potential trigger for the here reported long-lasting transcriptional changes on hypothalamic neuron populations. Further studies are needed to determine for how long the virus remains in the system and when/if the neurons return fully to the healthy control condition.

A subanalysis of Agrp[+]/Npy[+], Lepr[+], POMC[+], Hdc[+], Pmch[+], and Hcrt[+] neurons showed a particularly strong response of the POMC[+] neurons compared to the other neuronal subtypes. POMC[+] neurons have been shown before to be activated during inflammation in various models, a mechanism that is speculated to be important for development of sickness behaviours (*Jin et al., 2016*; *Kang et al., 2020*). This fits well with our observation that the *POMC* gene it self is strongly upregulated at 3 and 7dpi. Microglia, the resident immune cells of the brain, can have based on their activation state different immunological functions and properties. Interestingly, our results demonstrate the activation of pro-inflammatory genes in the microglia population, whereas anti-inflammatory transcripts are depleted. Further, general microglia signature genes (i.e, *Hexb*, *Cx3cr1* and *Selplg*) were found down-regulated at 3 and partly at 7 dpi, indicating a clear shift in microglia homeostasis at 3 dpi. This assumption is supported by the activation of genes contributing to cell adhesion, signalling and differentiation as well as synapse signalling. A pro-inflammatory microglia state (M1) is induced during an acute immune response and provide acute defence and repair mechanisms. For this, a tight control via constant cell-to-cell communication is necessary to avoid excessive long-term activation and eventual neurodegenerative consequences (*Rodríguez-Gómez et al., 2020*). In contrast, anti-inflammatory microglia (M2) are regulating the phagocytosis of pathogen and cell debris, inhibit inflammation and restore tissue homeostasis (*Franco and Fernández-Suárez, 2015*; *Jurga et al., 2020*). While our data does not support the presence of an acute inflammatory process in the hypothalamus, what we instead find is an increased alert state of the resident immune cells (*Figure 9*).

The most striking shift in gene expression in our dataset was detected in oligodendrocytes and astrocytes at 3 and 7 dpi, which caused the separation of the cell populations into distinct subpopulations at 7 dpi for both cell types. In oligodendrocytes, we detect a distinct expression of S*erum/glucocorticoid Regulated Kinase 1* (*Sgk1*) in the 7-dpi population. This protein belongs to the family of glucocorticoid receptors (GCR) and has been associated with several environmental pertubations, including its up-regulation in the CNS during stress or during fasting in obese individuals (*Avey et al., 2018*; *Hinds et al., 2017*; *Miyata et al., 2015*; *Nonogaki et al., 2006*). Further, the activation of SGK1 in oligodendrocytes is believed to lead to morphological changes, potentially caused by abnormal swelling of cytoplasm in oligodendrocytes or arborization of their processes (*Miyata et al., 2015*; *Miyata et al., 2011*). *Fkbp5*, encoding for FK506-binding protein 5, was another strongly induced transcript in oligodendrocytes at 7 dpi. It has previously been shown to be induced by glucocorticoids (*Hähle et al., 2019*) and involved in myelination in the prefrontal cortex (*Choi et al., 2021*). One of the main functions of oligodendrocytes is myelin formation around neuronal axons and the metabolic support of these to ensure neuronal signalling and information processing (*Lee et al., 2012*; *Moore*

*et al., 2020*; *Saab et al., 2016*). In addition, oligodendrocytes are involved in orchestrating stress responses mediated by GCRs. This is relevant for our model, as glucocorticoids are released following immune activation. This release usually occurs in a bi-phasic pattern, with an initial peak at 2–3 dpi caused by pro-inflammatory cytokines and a second peak at 7–8 dpi driven by T-cell-specific cytokines (*Hermann et al., 1994*; *Silverman et al., 2005*). Since the disease-specific subpopulation of oligodendrocytes, we detect at 7-dpi shows markers related to GC signalling, this sub-cluster is likely driven by T–cell specific cytokines.

For the astrocytes, one of the most pronounced discriminatory factors for the 7-dpi specific population was the expression of the *Leucine-Rich Repeat-Containing Protein 8* A (*Lrrc8a*, also *Swell1*, annotated as ENSMUSG00000099041) transcript. *Lrrc8a* has putative regulatory transmembrane transporter functions and is one essential subunit of volume-regulated anion channels (VRAC). Hence, it plays an important role in cell volume changes and regulate synaptic transmission via glutamate release from astrocytes (*Osei-Owusu et al., 2018*; *Yang et al., 2019*). The upregulation of Lrrc8a suggests changes in astrocytic morphology in the hypothalamus in our model, comparable to what has been reported in the hippocampus during peripheral IAV infections (*Hosseini et al., 2018*). However, the functional importance of this change in our study stays elusive and needs further investigation. In addition, we showed that the *zinc finger homeobox 3* (*Zfhx3*) transcript is specifically up-regulated in astrocytes at 7 dpi. Its genomic locus has been associated with body mass index and it is further known to influence circadian rhythm and sleep (*Balzani et al., 2016*; *Parsons et al., 2015*; *Turcot et al., 2018*). The latter two are believed to be also regulated via astrocytic released glutamate (*Brancaccio et al., 2017*). Further, we identified a distinctive expressed gene in a part of the astrocyte population - *aldehyde dehydrogenase 1 A1* (*Aldh1a1*), which is contributing to retinol metabolism. The inhibition of Aldh1a1 leads to reduced weight gain (*Haenisch et al., 2018*). *Aldh1a1* is known to mediate the synthesis of GABA in astrocytes (*Kwak et al., 2020*), which is then secreted by *Best1*-channels and GABA transporters (GAT, *Slc6a1* and *Slc6a11*) transporters (*Mederos and Perea, 2019*). These two transporters were found robustly expressed in the here analysed astrocyte population but not differentially expressed at any time point. The release of glutamate by astrocytes is regulated via the excitatory amino acid transporters (glutamate transporter 1; GLT-1, also known as *Slc1a2* and Glutamate Aspartate Transporter 1; GLAST, also known as *Slc1a3*) (*Delgado, 2013*), which we found highly expressed in the here analysed astrocytes (*Figure 7—figure supplement 1*). Additionally, we identified an up-regulation of genes of the oxidoreductase system. Such modulations of the antioxidant gene expression system has been shown to regulate microglia activation states (*Shih et al., 2006*), so what we see might be a regulatory feature enabling astrocytes to fine tune the transitions between microglia homeostatic and inflammatory states. Taken together, the data suggest activation of two different transport processes in astrocytes. The expression of VRAC might facilitate glutamine release by astrocytes which can be taken up by neurons and converted to glutamate or GABA (*McKenna and Ferreira, 2016*; *Waagepetersen et al., 2003*). This could be a supportive mechanism to maintain base level of neuronal signalling, which might be affected by the down-regulation of genes in the citric acid cycle in neurons and the accompanied loss of downstream products as neuropeptides (*Figure 9*). At the same time, the expression of transmembrane transporters might contribute to the swelling of astrocytes by the uptake of glutamate. Morphological changes in astrocytes are connected to appetite regulation by redirecting glutamate away from neurons and hence reducing glutamatergic signalling (*Delgado, 2013*). This might be a fundamental factor for sickness related weight loss and gain, categorizing the here identified 7-dpi astrocyte population as regulating neuroendocrine processes. Further studies are needed to understand and confirm the functional identity of the 7 dpi subclusters.

In summary, the data presented here shows how the cellular state of non-neuronal and neuronal cell populations in the hypothalamus change during peripheral IAV. Our data suggests the existence of tight co-regulation where one cell type affects the state of another (*Figure 9*). This connectivity on the one hand potentially ensures protection of the tissue by surveillant microglia but also counterbalances the microglia activity by astrocytic regulation of their antioxidant machinery. In addition, we found evidence that non-neuronal cell types like astrocytes and oligodendrocytes are potentially supporting neurons with their metabolic and signalling integrity, by enabling more nutrient transport as well as neuropeptide supply. Altogether, our study provides the first evidence of complex hypothalamic gene expression changes at the single cell level during a peripheral viral infection.

## Methods

### Animals and viruses

The experiments conducted in the study were approved by the Danish Animal Experimental Inspectorate (license number: 2020-15-0201-00585) in accordance with the Directive 2010/63/EU of the European Parliament and Council on the protection of animal used for scientific purposes. Female C57BL/6NTac mice were obtained from Taconic at 8–10 weeks old. Animals were randomly split into groups of 5, fed ad libitum and kept at a 12 h day/night cycle. Virus stocks (H1N1 A/California/07/2009) were maintained at Prof. Jan P. Christensen Lab, University of Copenhagen. Virus stocks were propagated in Madin-Barby Canine Kidney cells (MDCK). Cell cultures were infected at 0,01 MOI and harvested at 90–100% CPE after 2–3 days. Supernatant was centrifuged to clear it of cell debris and supernatant was stored at –80 °C for titration and infection of animals. Titration was done as previously described by Uddbäck and colleagues (*Uddbäck et al., 2016*).

### Viral infections and tissue collection

Ten-week-old female C57BL/6 N were inoculated intranasally with H1N1 A/California/07/2009 with a dose of $5 \times 10^5$ pfu in 30 µl per animal. Three days, 7 days, and 23 days post inoculation (dpi) a group of five animals was randomly selected and sacrificed by cervical dislocation between 1:00-3:00 pm to minimize confounding effects from circadian rhythm or sleep/wake phases. The lungs, brains, and olfactory bulbs were removed, frozen on dry ice and stored in at –80°C. The lungs and olfactory bulb were used to determine infection and the hypothalamus of the brain was used for snRNAseq.

### Viral RNA measurement by PCR

The olfactory bulbs and the right lung were homogenized using an ULTRA-Turrax T8.01 homogenizer in the RLT lysis buffer from the RNeasy mini and midi kit (Qiagen). The rest of the RNA extraction was performed according to the manufactured protocol from the RNeasy mini and midi kit to extract RNA from brain and lung tissue. The concentration and purity of RNA in the samples was measured by a Nanodrop spectrophotometer (NanoDrop 1000, Thermo Fisher Scientific, Delaware, USA). All the samples used in this experiment had a 1.7–2.1 260/280 nm ratio. Next, the samples were diluted to 1 µg total RNA in each well and cDNA was synthesized by reverse transcription using the QuantiTect Reverse Transcription Kit.

### Tissue preparation, nuclei isolation, and hashing of nuclei

Mouse brains tissue were sectioned at 300 µm in a cryostat [Leica CM3050S] to identify the lateral hypothalamus (bregma –0.94 to –1.34) for punching. The sections were punched five times, and the tissue was transferred to a cooled Dounce-homogenizer containing Nuclei EZ lysis buffer to homogenize the tissue. The samples were filtered through a 40 µm mini cell strainer followed by a 25%/29% iodixanol gradient to separate debris from the nuclei. To be able to distinguish samples later in post-sequencing analysis, the pellet was resuspended in FACS buffer containing sample-specific hash-tagged antibody recognizing a nucleus specific protein. A hashtag is an antibody with an oligonucleotide tail with PCR handle, a unique barcode and a poly A tail.

### FACS sorting and snRNA-sequencing

The sample were stained with DAPI (0.5 µg/ml) and purified using the Sony SH800S cell sorter. Whole nuclei were gated using the DAPI fluorescence at 450±25 nm, to avoid debris, and sorted directly into RT Reagent B (10 X genomics) and further processed according to the company's guide and sequenced using a NovaSeq 6000 (Illumina). Commercially available anti-nuclear pore complex proteins antibodies from BioLegends, Inc were used for hash-tagging. Per lane five different antibodies, to distinguish the different samples were used. Across the 5 sequencing lanes we used the same set of antibodies (TotalSeq-A0451, Cat. No. 682205; TotalSeq-A0452, Cat. No. 682207; TotalSeq-A0453, Cat. No. 682209; TotalSeq-A0454, Cat. No. 682211; TotalSeq-A0455, Cat. No. 682213).

### Single-nucleus RNA-sequencing raw data processing and quality control

Raw sequencing reads were processed using the 10 x Genomics Cell Ranger version 3.0 pipeline and aligned to the *Mus musculus* (mm10) genome with default parameters. All libraries were converted to

FASTQs using bcl2fastq. The FASTQ files were aligned to the mouse genome (assembly - GRCm38. p6) and subsequent read quantification was performed using the Salmon pipeline. Each library underwent separate quality control steps as described in the Alevin pipeline (*Srivastava, 2020*). Hashtagged antibody tag oligos (HTO) filtering removes nuclei with low tags or noisy cells. Furthermore, nuclei containing two or more cell barcodes (doublets) were removed.

Cells were tested for their mitochondrial RNA content and outliers with high mitochondrial RNA were removed using the quality control (QC) functions (perCellQCMetrics, PercentageFeatureSet, quickPerCellQC, isOutlier) from the R package *scuttle* (*McCarthy et al., 2017*). Counts were normalized using a log transformation of a genes counts divided by the total counts of a cell multiplied by 1e6. To remove variance attributed to batch and treatment effects for downstream visualization and clustering, samples were normalized with the 'SCTransform' method and integrated using the 'FindIntegrationAnchors' and 'IntegrateData' functions from Seurat (Version 4.1.0) (*Hao et al., 2021*). The integrated dataset was centred and scaled, PCA was carried out and for visualization we used UMAP dimensionality reduction.

## Single-nucleus RNA-sequencing cell population identification

Cells were clustered into neuronal and non-neuronal cells using a Gaussian mixture model (GMM) based on the expression of pan-neuronal markers (*Syp*, *Galntl6*, *Rbfox3*, *Syn1*, *Dnm1*). Cells in clusters with high expression of these markers were classified as neurons and low expression as non-neuronal cells. The dataset was separated based on this classification and subsequently analysed independently. For identifying GABAergic and glutamatergic cells we used likewise a GMM based on known GABAergic and glutamate markers (*Slc17a6*, *Slc32a1*, *Gad1*, *Gad2*). Cells in clusters with high expression of GABAergic markers and low expression of the glutamatergic marker were defined as GABAergic cells and vice versa cells in cluster with high expression of the glutamatergic marker and low expression of the GABAergic markers were defined as glutamatergic. We divided the dataset further based on this classification. We identified a neuronal cluster expressing neither of the GABAergic nor glutamatergic markers. It also did not express any other neuron markers (histaminergic (HA), dopaminergic, ect.), and the cluster was therefore moved to the non-neuronal dataset.

Cluster identification was performed on the integrated dataset using the Seurat functions 'FindNeighbors' and 'FindClusters using the Leiden algorithm. Non-neuronal cell populations and well known (i.e HCRT, HA, MCH, etc.) neuronal populations were identified and classified based of known marker genes. Additionally, we used published datasets from the hypothalamus (*Campbell et al., 2017*; *Chen et al., 2017*; *Mickelsen et al., 2019*; *Mickelsen et al., 2020*; *Romanov et al., 2017*; *Steuernagel et al., 2022*; *Zeisel et al., 2018*) to confirm cell type annotations by using the Seurat functions 'FindTransferAnchors' and 'TransferData'. We accepted a prediction score ≥ 0.5 in at least 50% of the cells of a cell cluster. Marker genes for the different cell populations were identified using the 'FindMarkers" function from the Seurat package and using the receiver operating characteristic (ROC) method.

## Differential gene expression analysis and case-control based expression shifts (Cacoa)

To calculate gene expression fold changes in samples from infected mice compared to the control group, we used a pseudo-bulk approach. We calculated in total three different differential gene expression datasets; (I) Aggregating counts per gene across all cells without discriminating between cell types or clusters, (II) aggregating counts per gene for all GABAergic, glutamatergic, and non-neuronal cells and (III) aggregating counts per gene for each identified cluster. Differential gene expression analysis was performed used the *DESeq2* and *scran* R packages. Calculations were performed per time point. In order to account for biological variance we selected only cell types which were comprised of nuclei from at least three animals. Furthermore, we wanted to assure to only include cell types with a sufficient amount of nuclei (at least 9 nuclei/time point) and only included a cell type if it contained at least three nuclei from at least three individual animals. Further we selected only robustly expressed genes in a cell cluster for DGE analysis, meaning a gene must be expressed in at least 95% of the cells of one of the tested time points versus the control to be included in the analysis. The counts were aggregated using the Seurat function 'AggregateExpression' and transformed to a DESeq2 data set with the function 'DESeqDataSetFromMatrix'. We calculated normalization factors using the scran

function 'computeSumFactors'. The differential expression analysis was performed using a likelihood ratio test (LRT) and a Gamma-Poisson Generalized linear model (glmGamPoi) for fitting the dispersion to the mean intensities. p Values were adjusted (for time points tested x number of cell populations x the number of genes) using the false discovery rate by Benjamini and Hochberg.

To dissect transcription responses for key neuronal cell populations (*Agrp/Npy+*, *Lepr+*, *Pomc+*, *Hcrt+*, *Pmch+*, and *Hdc+*) we performed and additional differential gene expression analysis with adjusted parameters to identify DEGs also for small cell clusters (*Supplementary file 6*). Here we accepted a cell cluster at a time point if it contained at least two cells from two individual animals.

We performed a case-control analysis based on the *Cacoa* package developed by Pethkhov et al. (*Petukhov et al., 2022*), comparing the different infected time points (cases) against the control group. Therefore, we separated the Seurat object with all data into three different sub-dataset containing the 3 dpi and control, 7 dpi and control as well as 23 dpi and control samples. We performed cluster-based density and expression shifts and cluster-free analysis on default settings.

## Gene set enrichment analysis

For gene ontology enrichment analysis (GOEA), we used the *topGO* package in R (*Alexa, 2022*). All significant differentially expressed genes (FDR ≤ 0.05) were defined as the gene universe. Only highly regulated genes (FDR ≤ 0.05, logFC >1 or logFC < −1) were included in the GOEA and tested twice, first implementing a classical algorithm, and second using an algorithm eliminating (*elim* method) local dependencies between GO terms. For the t-test statistic we used in both cases a Fisher exact test. p-Values returned by the *elim*ination method are considered as not affected by multiple testing and hence can be interpreted as corrected (*Alexa, 2022* ). Enrichment analysis for the Reactome database was performed with the 'ReactomePA' R package and its 'enrichPathway' function. We mapped the mice gene identifiers to human Entrez gene IDs for analysis. Genes were selected based on their expression in the same way as for GOEA. p Values were adjusted for multiple testing using the Benjamini & Hochberg correction method.

## Acknowledgements

This project was funded by the Independent Research Fund Denmark #9039-00132B (RL and BRK), the Lundbeck Foundation R344-2020-749 (CE) and by a Carlsberg Foundation Young Researcher Fellowship #CF19-0303 (NB and BRK). THP and KLE acknowledges the Novo Nordisk Foundation (unconditional donation to the Novo Nordisk Foundation Center for Basic Metabolic Research; grant number NNF18CC0034900) and the Lundbeck Foundation (Grant number R190-2014-3904). We acknowledge Helle Kinggaard Lilja-Fischer, Pernille Keller Andersen, and The Single-Cell Omics platform at the Novo Nordisk Foundation Center for Basic Metabolic Research (CBMR) for technical and computational expertise and support.

## Additional information

### Funding

| Funder | Grant reference number | Author |
|---|---|---|
| Independent Research Fund Denmark | 9039-00132B | René Lemcke<br>Birgitte R Kornum |
| Lundbeck Foundation | R344-2020-749 | Christine Egebjerg |
| Carlsberg Foundation | CF19-0303 | Nicolai T Berendtsen<br>Birgitte R Kornum |
| Novo Nordisk Foundation Center for Basic Metabolic Research | NNF18CC0034900 | Kristoffer L Egerod<br>Tune H Pers |
| Lundbeck Foundation | R190-2014-3904 | Kristoffer L Egerod<br>Tune H Pers |

| Funder | Grant reference number | Author |
|---|---|---|

The funders had no role in study design, data collection and interpretation, or the decision to submit the work for publication.

## Author contributions

René Lemcke, Conceptualization, Data curation, Formal analysis, Validation, Investigation, Visualization, Methodology, Writing – original draft, Writing – review and editing; Christine Egebjerg, Methodology, Writing – original draft, Writing – review and editing; Nicolai T Berendtsen, Kristoffer L Egerod, Methodology, Writing – review and editing; Allan R Thomsen, Conceptualization, Resources, Writing – review and editing; Tune H Pers, Resources, Software, Methodology, Writing – review and editing; Jan P Christensen, Conceptualization, Resources, Methodology, Writing – review and editing; Birgitte R Kornum, Conceptualization, Supervision, Funding acquisition, Investigation, Writing – original draft, Project administration, Writing – review and editing

## Author ORCIDs

René Lemcke (iD) http://orcid.org/0000-0002-4903-4741
Christine Egebjerg (iD) http://orcid.org/0009-0006-2035-6145
Tune H Pers (iD) http://orcid.org/0000-0003-0207-4831
Jan P Christensen (iD) https://orcid.org/0000-0002-4299-9479
Birgitte R Kornum (iD) https://orcid.org/0000-0002-2515-9451

## Ethics

The experiments conducted in the study were approved by the Danish Animal Experimental Inspectorate (license number: 2020-15-0201-00585) in accordance with the Directive 2010/63/EU of the European Parliament and Council on the protection of animal used for scientific purposes.

Reviewer #1 (Public Review): https://doi.org/10.7554/eLife.87515.3.sa1
Reviewer #2 (Public Review): https://doi.org/10.7554/eLife.87515.3.sa2
Author Response: https://doi.org/10.7554/eLife.87515.3.sa3

# Additional files

## Supplementary files

• Supplementary file 1. Number of cells in different cell populations across different cluster levels. Sheet QC counts: The dataset underwent different quality control steps. This tables contains the amount of cells per animal at the different control and filter steps. **Sheet Cell.counts_celltype.lev1**: Table contains the amount of cells for the different sequenced animals in neuronal or non-neuronal cells. **Sheet Cell.counts_celltype.lev2**: This table contains the amount of cells per anima across all GABAergic, glutamatergic or non-neuronal cells. **Sheet Cell.counts_cluster.id**: This table shows the number of cells per animal for the different identified cell clusters.

• Supplementary file 2. Results from cell label transfer from different datasets (see Methods) on cell cluster level. Containing information about the dataset, reference data ID, number of total cells, number of cells that fitted the prediction score criteria (score >0.5) and percent of cells in cell type tagged with selected prediction score cut-off. **Sheet Glut**: Results from the cell label transfer for glutamatergic clusters. **Sheet Gaba**: Results from the cell label transfer for GABAergic clusters. **Sheet NonN**: Results from the cell label transfer for non-neuronal clusters.

• Supplementary file 3. Predicted cell cluster annotations based on cell label transfer. Sheet Selection criteria: Describes the criteria for transferring cell labels to the identified clusters. **Sheet NonN**: Cell labels for non-neuronal clusters. **Sheet Glut**: Cell labels for glutamatergic clusters. **Sheet GABA**: Cell labels for GABAergic clusters.

• Supplementary file 4. Marker genes identified using a receiver operating characteristic (ROC) method. Sheet GABA: List of marker genes for GABAergic clusters. **Sheet Glut**: List of marker genes for glutamatergic clusters. **Sheet NonN:** List of marker genes for non-neuronal clusters.

• Supplementary file 5. Results from pseudobulb differential gene expression analysis. The analysis was performed 3 times on different cell type levels. **Sheet 3dpi - DEG - across all cells**: Differential expressed genes comparing all control cells vs all cells at 3dpi. **Sheet 7dpi - DEG - across all**

**cells**: Differential expressed genes comparing all control cells vs all cells at 7dpi. **Sheet 23dpi - DEG - across all cells**: Differential expressed genes comparing all control cells vs all cells at 23dpi. **Sheet GABA - DEG - across celltype**: Differential expressed genes from a pseudo-bulk analysis of GABAergic cells. **Sheet Glut - DEG - across celltype**: Differential expressed genes from a pseudo-bulk analysis of glutamatergic cells. **Sheet NonN - DEG - across celltype**: Differential expressed genes from a pseudo-bulk analysis of non-neuronal cells. **Sheet GABA - DEG - across cellcluster**: Differential expressed genes from a pseudo-bulk analysis on GABAergic cell cluster levels. **Sheet Glut - DEG - across cellcluster**: Differential expressed genes from a pseudo-bulk analysis on glutamatergic cell cluster levels. **Sheet NonN - DEG - across cellcluster**: Differential expressed genes from a pseudo-bulk analysis on non-neuronal cell cluster levels.

• Supplementary file 6. Results from pseudo-bulk differential gene expression analysis with less restrictive filter criteria on selected neuronal cell clusters (see Methods). Sheet 3dpi: Analysis results for comparison Control vs 3dpi. **Sheet 7dpi**: Analysis results for comparison Control vs 7dpi. **Sheet 23dpi**: Analysis results for comparison Control vs 23dpi.

• Supplementary file 7. Results from Reactome pathway enrichment analysis.

• Supplementary file 8. Results from Gene Ontology (GO) enrichment analysis.

• Supplementary file 9. Gene programs from cluster free expression shift analysis from Cacoa package. Analysis was performed for Control against the 3 different time points separately. Sheet Program scores UMAP embeddings: Contains the Adjusted z-scores from the identified genes programs at the different time points in all cells. **Sheet 3 dpi - gene programs**: Contains gene names, sim scores and loadings (for more information see *Petukhov et al., 2022*) for 9 different gene programs identified comparing cells from control samples against cells at 3 dpi. **Sheet 7 dpi - gene programs**: Contains gene names, sim scores and loadings (for more information see *Petukhov et al., 2022*) for 9 different gene programs identified comparing cells from control samples against cells at 7 dpi. **Sheet 23 dpi - gene programs**: Contains gene names, sim scores and loadings (for more information see *Petukhov et al., 2022*) for 8 different gene programs identified comparing cells from control samples against cells at 23 dpi.

• MDAR checklist

## Data availability

Raw data filtered count matrices and metadata are available on GEO under accession number: GSE226098.

The following dataset was generated:

| Author(s) | Year | Dataset title | Dataset URL | Database and Identifier |
|---|---|---|---|---|
| Kornum BR | 2023 | Molecular consequences of peripheral influenza a infection on cell populations in the murine hypothalamus | https://www.ncbi.nlm.nih.gov/geo/query/acc.cgi?acc=GSE226098 | NCBI Gene Expression Omnibus, GSE226098 |

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
