## [Editor Report · eLife assessment]

This **important** study combines experiments and computational approaches to understand the effects of influenza H1N1 infection on hypothalamic cells. The methodology and analysis are **solid** and raise questions around how a respiratory virus affects the central nervous system.

---

## [Referee Report · Reviewer #1 (Public Review)]

This study uses single-cell genomics and gene pathway analysis to characterize the transcriptional effects of influenza H1N1 infection on hypothalamic cell types. The authors use droplet-based single-nuclei RNA-seq to profile genome-wide RNA expression in adult mouse hypothalamic cells at 3, 7, and 23 days after intranasal infection with the H1N1 influenza virus. Through state-of-the-art and rigorous computational methods, the authors find that many hypothalamic cell types, glia, and especially neurons, are transcriptionally altered by respiratory infection with a non-neurotropic influenza virus and that these alterations can persist for weeks and potentially affect cell type interactions that disrupt function. For instance, microglia shift towards a pro-inflammatory molecular phenotype at 3 days post-infection, while astrocytes and oligodendrocytes significantly alter their expression of oxidoreductase activity genes and transport genes, respectively, at 7 days post-infection. In addition, POMC neurons of the arcuate hypothalamus, which suppress appetite and increase metabolism, appear to be unusually sensitive to H1N1 infection, upregulating more genes than other hypothalamic neurons. The authors' thorough discussion of the findings raises interesting questions and hypotheses about the functional implications of the molecular changes they observed, including the physiological changes that can persist long after acute viral infection. Given the role of the hypothalamus in homeostasis, this work sheds light on potential mechanisms by which the H1N1 virus can disrupt cell function and organismal homeostasis beyond the cells that it directly infects.

---

## [Referee Report · Reviewer #2 (Public Review)]

The new work from Lemcke et al. suggests that the infection with Influenza A virus causes such flu symptoms as sleepiness and loss of appetite through the direct action on the responsible brain region, the hypothalamus. To test this idea, the authors performed singothalamus in controlled experimental conditions (0, 3, 7, and 23 days after intranasal infection) and analyzed changes in the gene expression in the specific cell populations. The key results are promising and spurring future research. After revision, the analysis was considerably improved. Alternative approaches were used for testing. Specifically, during the revision: (1) The annotation of cell types was considerably improved; (2) The authors performed an additional analysis comparing case-control studies (Cacoa), where they could partly confirm their earlier findings.

---

## [Author Response]

The following is the authors’ response to the current reviews.

**Reviewer #1 (Recommendations For The Authors):**
The revision and rebuttal have addressed all concerns raised in the initial review. Upon review of the revised figures, however, it is unclear why Figure 8C shows many significant DEGs in POMC neurons (which according to Figure 8b is the "GABA_24" cluster), whereas Figure 6A shows few to no DEGs in the GABA_24 cluster. Same for Pmch neurons/Glut_25, which seem to be missing from Figure 6A.

Answer: In order to capture changes in these smaller cell population we performed an additional DEG analysis with modified and less strict parameters (compared to the first main analysis). We mention the different parameters in the methods part of the revised manuscript (Differential gene expression analysis and case-control based expression shifts (Cacoa)).

The following is the authors’ response to the original reviews.

**Reviewer #1 (Recommendations For The Authors):**
Major issues1. A key conclusion of this study is that neurons show longer lasting infection-related changes in gene expression than do non-neuronal cells, suggesting that neurons are more persistently affected, which could potentially underlie persistent effects of infection on behavior or physiology. However, the authors also report that over twice as many transcripts were captured in neurons than in non-neuronal cells, and that neurons and non-neurons were not equal in number. The number of transcripts and cells per cell type can affect the likelihood of detecting a differentially expressed gene when comparing cell types. Thus, the difference in infection related DEGs between non-neuronal cells and neurons may be due in part to differences in the numbers of transcripts and cells in each group. How would the number of infection related DEG's compare if the same number of transcripts were detected in neurons as in non-neuronal cells? In addition, is there any relationship between thenumber of infection related DEGs detected and the number of cells in the respective groups?

We performed an additional analysis, down sampling the transcripts per cells to similar numbers (~1600 transcripts/cell), showing a similar pattern as shown in the original calculation of DEGs. High downregulation of genes in GABAergic, Glutamatergic and Nonneuronal cells at 3 and 7 dpi, but long-lasting dis-regulation at 23 dpi only in the neuronal subtypes. The analysis results can be found in Supplementary figure12 and on page 11 in the results section.

1. The rationale for focusing on the LH and DMH is unclear. While these regions do play important roles in control of body weight and wakefulness, the authors do not report whether the cell types relevant to these functions are among those affected by infection. For instance, the authors mention HCRT and MCH neurons in the introduction but do not comment on whether these neurons show any significant changes after H1N1 infection in their analysis. Also, what about the POMC neurons or the Lepr+ DMH neurons? Knowing whether and how these body weight associated cell types are affected could help to connect the phenotypic (e.g., body weight) and molecular changes observed.

We have added an additional analysis of some well know hypothalamic subtypes. What is interesting is that the different neuronal subtypes respond to the infection differently. While most neurons show the strongest response at 3dpi, POMC+ neurons show consistent changes across all three time points. This could point to different neuronal subtypes paying different roles in the sickness response to the influenza infection. The new data has been added to Figure 8 together with new text in the result section and discussion (Page 17 & 20).

1. For discriminating neurons and non-neuronal cells based on their expression of neuronal marker genes, was this performed at the single-cell level or the cluster level? Similarly, was the discrimination of GABAergic and glutamatergic neurons done at the cell or cluster level?

The discrimination of the cell types was done on single cell level. This information has been added to the revised manuscript on page 25.

1. The authors mention that body weight did not change in some of the mice. Was there any difference in infection related DEGs between the mice that lost weight and the mice that didn't? Was there any correlation between the molecular and phenotypic (i.e., body weight) changes observed?

We agree that this could have been an interesting point to investigate, however, we can only say with certainty for 2 animals in the recovery group (23.7 and 23.8) that they didn’t lose weight (Supplement figure 2). In Figure 4A we show that overall the different time points group well together, with exception for animal 23.7 which seems to have a better overlap with 7 dpi, indicating that we possibly captured here a delayed disease response. However, to make any indepth analysis, we have to few animals without weight-loss.

5)The authors noted that the hypothalamic neurons continue to show infection-related changes in gene expression at 23dpi though body weight has returned to normal. In this H1N1 model, are there any persistent behavioral deficits at 23dpi that could be explained by the persistent changes in gene expression in DMH and LH neurons?

We did not test for long-lasting behavioural changes in these animals. Another study by Hosseini et al. (https://www.ncbi.nlm.nih.gov/pmc/articles/PMC6596076/) focus on cognitive long term effects of viral infections. Even though they did not include the here used H1N1 model, they included the PR8 strain, but didn’t report any long lasting behavioural or cognitive changes. So far only cognitive deficits during the acute phase of the infection caused by the PR8 H1N1 model have been shown. This would be a very interesting follow up study to perform, but this, we believe, is out of scope for the current manuscript.

1. In Figure 1F, the 3dpi sample appears to differ from the other samples in terms of its neuron/non-neuron composition. The authors point this out but offer no discussion or further analysis. Was this difference driven by one or more cell types? Is this difference likely to be technical (e.g., less white matter in sample = fewer oligodendrocytes), or could this be related to the infection (e.g., glial death or neurogenesis at 3dpi)?

We have added the location of the punching within the hypothalamus for the different groups to the supplements (Supplementary Figure 3). The differences in neuron/non-neuron composition could originate from differences in the punching location, but we do not have data to support this conclusion. The difference could also stem from biological alterations during the infection.

1. Since influenza viruses replicate in the cell nuclei, did the authors capture any H1N1 RNA in their single-nuclei RNA-seq samples?

We mapped the single nuclei data against the viral genes, but could not detect any of the viral genes in the data set. We are still optimizing detecting of low amounts of viral genes in snRNA-seq data and have not included this information in the manuscript. We believe, that the virus did not manage to migrate in the hypothalamus and infiltrate the cells in the here captured area.

Minor Issues1. Page 1. The abstract ends with the sentence: This is complemented by increased activity of microglia monitoring their surroundings. Presumably, the authors are basing this statement on the functions of genes altered in microglia by infection. However, saying that microglia behavior has changed is a bit of a stretch here, since the results suggest a change in the molecular phenotype of microglia but do not demonstrate a change in their behavior.

We agree that the phrasing of the end of the abstract was not accurate and didn’t reflect the outcome of the analysis. We adjusted the sentence to: “The change of microglia gene activity suggest that this is complemented by a shift in microglia activity to provide increased surveillance of their surroundings.” Which should provide a better idea that the findings we present are a suggestion based on the transcriptomic changes in the cell population (Page 1).

1. Page 8. The authors refer to Th+, Ddc+ neurons as dopaminergic. However, adrenergic/noradrenergic neurons also express these genes. How do the authors know the neurons are not adrenergic/noradrenergic?

There are to our knowledge no nor-adrenaline/adrenaline producing neurons in the hypothalamus. In contrast dopaminergic neurons have indeed been identified in this area.

1. In the Methods section, Slc17a6 and Slc32a1 are not "pan-neuronal markers" since they are only expressed by subsets of neurons.

We removed the glutamatergic and GABAergic marker genes (Slc17a6 and Slc32a1) from the list of neuronal markers. They are stated further down in the method section as glutamatergic and GABAergic markers. Find the changes on (Page 24/25).

1. Was the hashtagging antibody custom or commercial? If commercial, what was the source, catalog #, lot #? If custom, the authors should describe how it was made and validated.

We used commercial antibodies for hash-tagging. We added the missing information to the manuscript and can be found on Page 24 of the revised manuscript.

1. In the data processing section of the Methods, SCTransform is mentioned twice. Was normalization with SCTransform applied twice?

The data was only normalized once using the SCTransfrom method. We adjusted the part of the method section to make it more clear (Page 24).

1. In the section on gene set enrichment analysis, the first sentence includes this text: "(is a reference needed?)." The answer is yes - Alexa A, Rahnenfuhrer J (2022). topGO: Enrichment Analysis for Gene Ontology. R package version 2.50.0.

The missing reference was added (Page 26).

1. Page 4: "leaved" should be corrected to "left"

The wrong wording was corrected.

1. Figure 2D - gene is labeled as Slc31a1 on the figure and Slc32a1 in the figure legend

We provided a new Figure plate with the right marker genes.

1. Official gene IDs should be italicized

We checked the gene IDs again, and italicized wrongly formatted gene IDs.

1. It is not clear whether the authors are planning to share their code. However, their code would be needed to reproduce their results, since the methods section provides a summary of what was done but lacks key details (e.g., parameters and software packages used during data processing and analysis)

Code will be shared on request. We added this also to the revised manuscript (Page 1).

**Reviewer #2 (Public Review):**

The new work from Lemcke et al suggests that the infection with Influenza A virus causes such flu symptoms as sleepiness and loss of appetite through the direct action on the responsible brain region, the hypothalamus. To test this idea, the authors performed single-nucleus RNA sequencing of the mouse hypothalamus in controlled experimental conditions (0, 3, 7, and 23 days after intranasal infection) and analyzed changes in the gene expression in the specific cell populations. The key results are promising.However, the analysis (cell type annotation, integration, group comparison) is not optimal and incomplete and, therefore should be significantly improved.More specifically:1. The current annotation of cell types (especially neuronal but also applicable to the group of heterogeneous "Unassigned cells") did not make a good link to existing cell heterogeneity in the hypothalamus identified with scRNA seq in about 20 recently published works. All information about different peptidergic groups can not be extracted from the current version (except for a few). There are also some mistakes or wrong interpretations (eg, authors assigned hypothalamic dopamine cells to the glutamatergic group, which is not true). This state is feasible to improve (and should be improved) with already existing data.

We repeated the cell label transfer with the newly published HypoMap and added additional information to the supplements about the cell type assignments. Additionally, we agree that the dopaminergic neurons do not belong to the group of glutamatergic neurons, however assigned them into this group based on the clustering. We changed the phrasing in the results, to make a better differentiation between the two groups (Page 8).

1. I am confused with the results shown in the label transfer (suppl fig 3 and 4; note, they do not have the references in the text) applied to some published datasets (authors used the Seurat functions 'FindTransferAnchors' and 'TransferData'). The final results don't make sense: while the dataset for the arcuate nucleus (Campbel et al) well covered the GABAergic neurons it is not the case for the whole hypothalamus datasets (Chen et al; Zeisel et al). Similarly, for glutamatergic neurons. Additionally, I could not see that the label transfer works well for PMCH cells which should be present in the dataset for the lateral hypothalamus (Mickelsen et al,2019).

We performed the additional label transfer of the hypothalamus data. Here we accepted a prediction score of 0.5 and transferred a cell type label to our annotated cluster IDs, if at least 10% of cells within a cluster were annotated with the 0.5 prediction score. We found that well defined neuron population types like Hcrt+, Pmch+ and Hdc+ neurons as well as Pomc+ neurons were tagged with a high predictions scores ( >= 0.9, Supplement Figures 6 and 7) and non-neuronal cell types (Supplement Figure 8) were well annotated. Additionally we identified an Agrp+ neuron population with the Gaba_1 neurons. This information has been added to the revised manuscript (Pages 6, 8).

1. There are newly developed approaches to check the shifts in the cell compositions and specific differential gene expression in the cell groups (e.g. Cacoa from Kharchenko lab, scCoda from Büttner et al; etc). Therefore, I did not fully understand why here the authors used the pseudo-bulk approaches for the data analysis (having such a valuable dataset with multiple hashed samples for each timepoint). Therefore it would be great to use at least one of those approaches, which were developed specifically for the scRNAseq data analysis. Or, if there are some reasons - the authors should argue why their approach is optimal

We performed an additional analysis comparing case-control studies (Cacoa). We perfomed both modalities, cluster-based and cluster-free expression shifts and cell type compositions We could partly confirm our findings using the pseudo-bulk approach. The clusterspecific density shift (Supplement Figure 15) identified only shifts in non-neuronal cell types between the Control group and 3 dpi. We believe, these composition shifts are caused by the lower number of non-neuronal cells in the 3 dpi time point. Cluster-specific expression shifts show similar results as in the pseudo-bulk approach, with significant expression shift identified at 3 and 7 dpi in neuronal and non-neuronal cell clusters (Supplement Figure 16). However, no significant expression shifts were identified in the recovery group at 23 dpi. Using the cluster-free expression shift approach, however we were able to identify a similar picture as described with the pseudo-bulk approach. In the recovery group at 23 dpi, we found mainly changed gene programs in neuronal cells, and no transcriptional changes in the non-neuronal cells (Supplement Figure 17-20). This new analysis has been added to the revised manuscript (Pages 4-6, 26) including supplementary figure and tables as stated.

1. When the authors describe the DGE changes upon experimental conditions (Figures 5 and 6), my first comment is again relevant: it is difficult to use the current annotation and cell type description as the reference for testing virus effects and shifts in the DGE in distinct neuronal subtypes.

The cell type annotations have been checked and additional label transfer has been performed. All figures in the manuscript has been updated.

I have to note that the experimental design is well done and logical. Therefore I believe that to strengthen the conclusions, the already obtained datasets can be used for improved analysis.

**Reviewer #2 (Recommendations For The Authors):**
I have some minor concerns:1. For the quality check it would be good to see how different hashed samples for each timepoint cover the UMAP embeddings.

We added the UMAP embeddings to the supplement (Supplement Figure 4).

1. In Fig 1e colors are not optimal - it is impossible to assess it.

We separated the UMAPs for the different time points to make it easier to assess. See updated Figure 1E.

1. In the methods authors started "Single-nucleus RNA-sequencing cell population identification" from the description of using a Gaussian mixture model (GMM). However, I could not clearly understand how this model was used and which kind of result it provided.

We used an GMM model with known markers for neurons and in a second step for glutamatergic and GABAergic cells to sub-cluster the cells and then selected based on high and low expression of the marker genes in the cluster into their respective classes. This information has been added to the method section (Page 24/25).

1. Could the authors better clarify why "they calculated normalization factors using the scran function 'computeSumFactors'" when working with pseudobulk analysis?

This size factor normalization was recommended for single cell data by the authors of the DESeq2 packages.

http://bioconductor.org/packages/devel/bioc/vignettes/DESeq2/inst/doc/DESeq2.html

1. I didn't find logic in "a cell cluster was only included if it contained more than 2 nuclei in at least 3 individual animals" (page 24). Maybe I misinterpreted it.

The rationale for the selection methods was based on the findings that not all animals in the recovery group had the same effects in weight loss. The acute time points didn’t show enough weight loss to decide if all animals in these groups lost the same amount and were equally sick. Hence, in order to have biological robustness we decided to only analyse clusters where cells from at least 3 animals at a specific time point contributed to a cell type. In order to have enough cells per cell type for the calculation of DEGs, we decided to only include a cell type at a specific time point if it contained at least 3 cells from one individual. This selection method limits the analysis to cell types with at least 9 cells per time point.